# *Evil in the Pairing Assumption*: MULTIMODAL ATTRIBUTION VIA ADAPTIVE INFORMATION BOTTLENECK

## ABSTRACT

Multimodal attribution methods such as M2IB aim to interpret vision-language models without requiring task-specific labels, but they often rely on the assumption of accurate semantic alignment between image-text pairs. This assumption does not hold in open-world settings, where noisy or mismatched inputs are common. Under such conditions, existing attribution methods tend to overfit and generate forced explanations, compromising the reliability and trustworthiness of interpretability. To address this, we observe that a well-balanced trade-off between the compression and prediction terms in the information bottleneck objective can mitigate overfitting. Based on this insight, we introduce an attribution framework that leverages an adaptive information bottleneck optimisation objective. Our method dynamically adjusts the bottleneck constraints without assuming reliable cross-modal alignment. Extensive experiments on large-scale image-text datasets show that our approach consistently outperforms existing attribution methods in both quantitative metrics and qualitative interpretability, providing more robust and trustworthy explanations while relaxing the requirement for aligned image-text pairs.

## 1 INTRODUCTION

Multimodal learning, particularly in vision-language models (VLMs), has made remarkable progress in recent years, enabling powerful capabilities across tasks such as image captioning, visual question answering, and retrieval. However, as these models become increasingly complex and ubiquitous, the demand for interpretable and trustworthy explanations has grown accordingly. Multimodal attribution methods aim to fill this gap by identifying which parts of the input (e.g. image regions or text tokens) contribute most to a model's decision. Among these, recent works such as the Multi-Modal Information Bottleneck (M2IB) (Wang et al., 2023) and the Narrowing Information Bottleneck (NIB) (Zhu et al., 2025) propose to utilise information bottleneck to generate faithful and compact explanations without task-specific supervision.

However, a critical assumption underlying these methods is the semantic alignment between paired modalities—that is, the image and text pairs are assumed to share the same concept or scene. While this assumption holds in well-curated datasets, it often breaks down in open-world scenarios, where noisy, mismatched, or loosely related pairs are common. Under such conditions, existing attribution methods tend to overfit to spurious correlations, resulting in misleading or uninformative explanations, as illustrated in Fig. 1. This undermines the interpretability and usability in real-world applications. These challenges highlight the need for attribution methods that can adapt to varying degrees of cross-modal alignment and remain robust in the presence of noise and semantic mismatch.

In this work, we challenge the pairing assumption and ask: *Can we build robust attribution methods that remain effective even when input modalities are weakly aligned or even completely mismatched?* To this end, we introduce a new framework that adapts the standard modality-matching objective in information bottleneck attribution. We do not assume clear alignment between modalities and propose Adaptive Multimodal Information Bottleneck (AdaIB), which extends the traditional Information Bottleneck (IB) objective by introducing an adaptive weighting mechanism between the compression term and the fitting term on the objective without requiring strict one-to-one alignment. Our key contributions are summarised as follows:

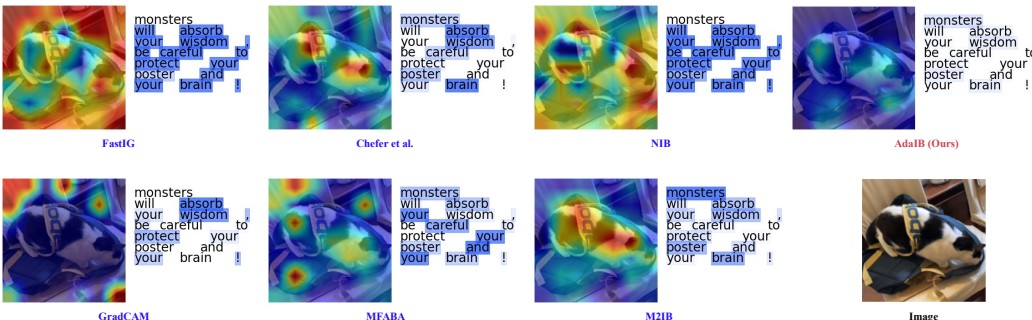

Figure 1: Visualisation results of various multimodal explanation methods on *completely noisy image-text pairs*. The example caption is "monsters will absorb your wisdom, be careful to protect your poster and your brain!". Existing approaches (e.g. M2IB, GradCAM, NIB, etc.) tend to produce forced or misleading alignments even when the modalities do not semantically match (First three columns). In contrast, our proposed AdaIB method is able to suppress responses under a completely mismatched pair (Top of the last column).

- We propose an adaptive information bottleneck optimisation framework (AdaIB) that dynamically adjusts the objective based on the relationship between paired image-text data, improving the attribution method for interpretability analysis.

- We formally analyse the functional properties of the proposed AdaIB, including its gradient behaviour and information ratio structure. We show that AdaIB enables sample-aware control overfitting and compression terms and recovers the standard IB as a special case.

- We empirically validate our approach across diverse metric conditions, showing much better robustness and interpretability compared to state-of-the-art methods.

## 2 RELATED WORK

### 2.1 UNIMODAL INTERPRETABILITY METHODS

Traditional interpretability methods were primarily developed for unimodal deep learning models and face significant limitations when applied to multimodal architectures. Early gradient-based approaches, such as Saliency Maps, compute input-output gradients to highlight important regions in the input. However, they are highly sensitive to noise and often produce low-resolution or unstable explanations. Grad-CAM Selvaraju et al. (2017) improves upon this by using gradients of class-specific activation maps in convolutional layers, resulting in more focused and human-interpretable heatmaps. LIME (Ribeiro et al., 2016) adopts a black-box approach by perturbing the input and fitting a local surrogate model, offering model-agnostic explanations. Despite its generality, LIME may fail on complex models due to its reliance on local linearity assumptions. RISE (Petsiuk et al., 2018) also introduces a model-agnostic strategy by applying random masks and measuring output changes to build relevance maps. While effective globally, its sampling-based nature leads to high computational cost and potential noise. More recent attribution methods, such as AGI (Pan et al., 2021) and MFABA (Zhu et al., 2024), utilise adversarial perturbations to generate more robust saliency maps. These approaches satisfy formal interpretability criteria like Sensitivity and Implementation Invariance (Sundararajan et al., 2017), offering theoretical guarantees.

Nevertheless, most of these methods are designed for unimodal tasks and require access to downstream task labels or internal gradients, making them less suitable for large-scale vision-language models. Attempts to extend such methods to the multimodal setting—such as CLIP—face challenges due to architectural differences and lack of task-specific supervision. As demonstrated in our experiments, these methods often fail to provide meaningful explanations when applied directly to multimodal models.

## 2.2 Multimodal Interpretability Methods

CLIP (Radford et al., 2021) learns joint image-text representations by training separate encoders on large-scale image-text pairs, aligning the two modalities in a shared embedding space. This design enables zero-shot transfer, allowing the model to perform various tasks based solely on natural language prompts, without requiring task-specific annotations. However, the complexity of multimodal reasoning raises the need for interpretability to ensure that the model's predictions are based on semantically meaningful features. Investigating CLIP's interpretability is thus crucial to determine whether it captures genuine vision-language associations or merely exploits spurious dataset correlations.

A range of methods have been proposed to interpret vision-language pre-trained models, yet many introduce limitations in fidelity, scalability, or practicality. M2IB (Wang et al., 2023) applies a multimodal information bottleneck to filter irrelevant features, but increases architectural complexity. COCOA(Lin et al., 2022) modifies Integrated Gradients with contrastive learning, requiring additional positive and negative samples that may introduce irrelevant context. TEXTSPAN (Gandelsman et al., 2023) and (Hossain et al.) rely on constructing sample-specific sets or selecting neighbours in embedding space, making them dependent on external data and less generalizable. LICO (Lei et al., 2024) retrains models to maintain cross-modal alignment, but its explanations apply to the altered model, not the original one, and are affected by training randomness. FALCON (Kalibhat et al., 2023) explains individual features via highly activating examples, but lacks per-instance interpretability.

These existing interpretability methods for vision-language models often rely on strong assumptions, such as clean image-text alignment, access to contrastive examples, or task-specific supervision. Many of these approaches require sampling additional inputs, retraining surrogate models, or modifying the model architecture—factors that reduce their robustness and practicality in open-world settings. In particular, when image-text pairs are noisy or entirely mismatched, these methods often still generate forced explanations, undermining the reliability and trustworthiness of the interpretability results, as shown in the Fig. 1. In contrast, our method avoids these limitations by introducing an adaptive information bottleneck objective that does not assume reliable modality alignment. Instead of enforcing a strict pairwise correspondence between images and text, we use a soft alignment mechanism that adapts to different degrees of semantic consistency, dynamically adjusting the optimisation of the compression and fitting terms in the information bottleneck theory for each image-text pair, resulting in more effective and credible interpretability.

# 3 The Information Bottleneck Principle In Multi-modal Interpretability

In this section, we provide a detailed explanation of how the Information Bottleneck (IB) principle can be applied to multimodal interpretability. We further identify key limitations arising from the optimisation objective of IB in the multimodal setting, which motivate the design of our proposed method.

The Information Bottleneck (IB) principle offers a theoretically grounded framework for interpretability by balancing the trade-off between compression and relevance. Specifically, it seeks to encode an input variable $X$ into a latent representation $Z$ that preserves maximal information about a target variable $Y$, while minimising the information retained about $X$ itself.

In the context of multimodal interpretability, this framework can be naturally extended by considering $X$ and $Y$ as different modalities, such as text and image. For text-to-image attribution, the textual input $x_T$ is compressed while maximising the mutual information $I(Z; x_I)$ with the corresponding image $x_I$. Conversely, for image-to-text attribution, the image $x_I$ is compressed with the objective of retaining information relevant to the text $x_T$. Under this formulation, attribution maps can be derived by assessing the extent to which each part of the input survives the bottleneck. **Dimensions that undergo strong compression (i.e. contribute less to $I(Z; Y)$) are deemed less informative for cross-modal alignment, whereas those that retain more information are considered more relevant.** Thus, the mutual information term $I(Z; Y)$ not only enforces semantic alignment between modalities but also provides a principled measure of feature importance based on information retention, without relying on gradient-based or perturbation-based explanations.

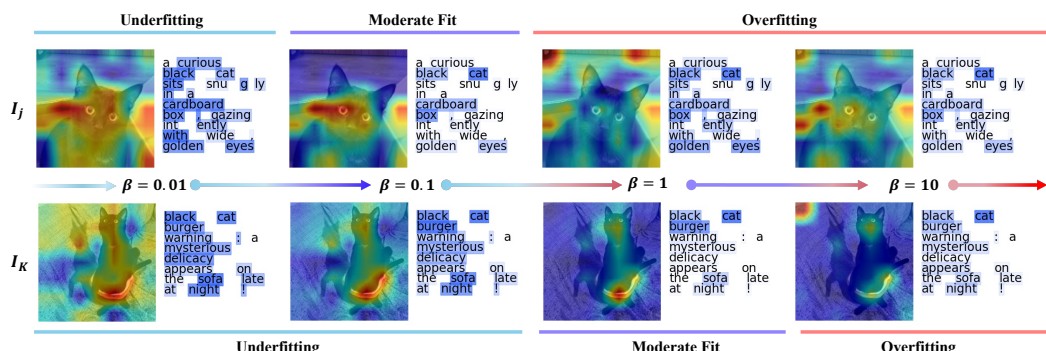

Figure 2: Varying $\beta$ in the Information Bottleneck controls the trade-off between compression and fitting. As the optimal $\beta$ for achieving good multimodal interpretability differs across samples (e.g. the 'Moderate Fit' for $I_j$ and $I_k$), a fixed $\beta$ is suboptimal. This diversity in optimal $\beta$ directly challenges the common practice in prior work, where $\beta$ is treated as a fixed hyperparameter for all samples. Our observation shows that this one-size-fits-all assumption is inappropriate. This observation motivates us to explore a dynamic adjustment of the IB trade-off parameter $\beta$ based on the semantic relationship between the input image-text pair.

As shown in Fig. 2, this suggests, 1) **an inappropriate choice of $\beta$ can lead to either overfitting or underfitting in the explanation process, resulting in overly specific or overly vague interpretations**. 2) **The optimal $\beta$ varies across different images; an adaptive, image-specific selection is essential for generating reliable explanations**. Based on this motivation, we design our method, which will be introduced in the following section.

## 4 METHOD

In this section, we introduce the *Adaptive Information Bottleneck (AdaIB)*, a novel framework that provides a sample-specific trade-off between sufficiency and representation compression. We first present the variational inference formulation that makes AdaIB tractable for deep learning optimisation. We then establish theoretical guarantees showing how AdaIB dynamically balances the two core principles of the Information Bottleneck (IB) theory—sufficiency and minimality.

### 4.1 ADAPTIVE INFORMATION BOTTLENECK (ADAIB)

The Information Bottleneck (IB) principle (Tishby et al., 2000) formulates representation learning as a trade-off between predictive sufficiency and input compression:

$$\mathcal{L}_{\text{IB}} = I(Z; Y) - \beta \cdot I(Z; X), \tag{1}$$

where $X$ denotes the input, $Y$ the target, and $Z$ a latent representation of $X$. The hyperparameter $\beta > 0$ controls the compression strength. The first term enforces that $Z$ retains task-relevant information about $Y$, while the second term limits the amount of information $Z$ carries from $X$.

Despite its elegance, the IB objective has two key limitations: (i) the coefficient $\beta$ is fixed across all samples, preventing adaptation to the varying relevance of data between $X$ and $Y$; and (ii) in noisy or multimodal settings, a fixed $\beta$ can result in underfitting of useful features or overfitting to irrelevant signals.

To address these issues, we propose the *Adaptive Information Bottleneck (AdaIB)*.

**Definition 1 (adaptive information bottleneck objective)** *Given an input $X$, a target $Y$, and a latent representation $Z$, the AdaIB objective is defined as*

$$\mathcal{L}_{\text{AdaIB}} = f(X, Y) \cdot I(Z; Y) \ - \ g(f(X, Y)) \cdot I(Z; X), \tag{2}$$

*where $f : \mathcal{X} \times \mathcal{Y} \to (0, \infty)$ is a relevance function that quantifies the statistical dependence between $X$ and $Y$, and $g : (0, \infty) \to (0, \infty)$ is a monotone non-increasing function that assigns the corresponding compression weight.*

In AdaIB, the relevance score $f(X, Y)$ adaptively balances sufficiency and minimality. Large $f$ emphasizes *sufficiency* by scaling up $I(Z; Y)$, while small $f$ emphasizes *minimality* via $g(f)$. Unlike classical IB with fixed $\beta$, AdaIB allows a sample-specific trade-off. See Section 4.3 for further discussion.

**Proposition 1 (classical IB as a special case)** *For any relevance value $f(X, Y) > 0$, define the effective coefficient $\beta_{\mathrm{eff}}(t) := g(f)/f$. Then in $(X, Y)$, the AdaIB objective rewrites exactly as*

$$\mathcal{L}_{\mathrm{AdaIB}} = f(X, Y) \Big( I(Z; Y) - \beta_{\mathrm{eff}}\big(f(X, Y)\big) I(Z; X) \Big).$$

*In particular, if $f \equiv c > 0$ and $g \equiv \beta c$ are constant across samples, then $\beta_{\mathrm{eff}} \equiv \beta$ and AdaIB reduces to the classical IB objective.*

Proposition 1 above shows that AdaIB is a sample-wise reweighted IB: $f(X, Y)$ acts as an importance weight on each pair $(x, y)$, while the trade-off parameter $\beta_{\mathrm{eff}}(f)$ adapts with relevance. Hence, unless $f$ and $\beta_{\mathrm{eff}}$ are constant, AdaIB is a strict generalisation of IB.

## 4.2 VARIATIONAL OBJECTIVE AND OPTIMISATION FOR ADAIB

The objective in Definition 1 contains mutual-information terms that are intractable in general, as in (Tishby et al., 2000). We derive a tractable variational formulation and an empirical training loss.

**Variational objective.** Using a standard lower bound for $I(Z; Y)$ and a variational substitution for $I(Z; X)$, we obtain

$$\mathcal{L}_{\mathrm{AdaIB}}^{\mathrm{var}} = \mathbb{E}_{p(x,y)}\Big[ f(X, Y)\, \mathbb{E}_{p(z|x)} \log q(y|z) \ - \ g\big(f(X, Y)\big) \, \mathrm{KL}\big(p(z|x) \,\|\, r(z)\big) \Big], \qquad (3)$$

where $q(y|z)$ and $r(z)$ are variational distributions. The additive term $H(Y)\, \mathbb{E}_{p(x,y)}[f(X, Y)]$ is constant w.r.t. $q, p, r$ and is dropped during optimization. Full derivations are provided in Appendix B.1.

**Empirical objective.** We estimate the expectations via Monte Carlo with the reparameterization trick, $z_i \sim p_\psi(z|x_i)$, which yields

$$\hat{\mathcal{L}} = \frac{1}{N} \sum_{i=1}^{N} \Big[ f(x_i, y_i)\, \log q(y_i|z_i) \ - \ g\big(f(x_i, y_i)\big) \, \mathrm{KL}\big(p(z|x_i) \,\|\, r(z)\big) \Big]. \qquad (4)$$

This estimator is sample-specific, end-to-end differentiable, and avoids explicit MI estimation.

## 4.3 SUFFICIENCY AND MINIMALITY BALANCE

In the classical Information Bottleneck (IB) theory, the two central desiderata of a representation are *sufficiency* and *minimality*. Sufficiency requires that the learned representation $Z$ preserves all task-relevant information about the target $Y$. Minimality requires that $Z$ discards all task-irrelevant information from the input $X$.

The Information Bottleneck principle aims to learn representations that are both *sufficient* and *minimal*. The adaptive relevance function $f(X, Y)$ promotes sufficiency by amplifying the fitting term when $X$ and $Y$ are strongly correlated, while the inverse weighting $g(f(X, Y))$ enforces minimality by strengthening compression when $X$ and $Y$ are weakly correlated. In what follows, we establish 3 theoretical properties of AdaIB:

(i) sufficiency when $f$ is large (Theorem 1);
(ii) minimality when $f$ is small (Theorem 2);
(iii) an adaptive trade-off balancing the sufficiency and minimality (Theorem 3).

**Theorem 1 (sufficiency at high relevance)** *Let $f : \mathcal{X} \times \mathcal{Y} \to (0, \infty)$ and $g : (0, \infty) \to (0, \infty)$ be locally Lipschitz with $g$ non-increasing. Assume that for the considered class of representations $Z$, the mutual informations satisfy $I(Z; Y) < \infty$ and $I(Z; X) < \infty$. Then as $f(X, Y) \to \infty$, the AdaIB objective satisfies*

$$\mathcal{L}_{\mathrm{AdaIB}} \ \sim \ f(X, Y) \cdot I(Z; Y),$$

*i.e.* $\mathcal{L}_{\text{AdaIB}}/(f(X,Y)\,I(Z;Y)) \to 1$. *Consequently, the optimisation places dominant emphasis on maximising $I(Z;Y)$, driving $Z$ toward sufficiency.*

Theorem 1 is proved in B.2 Proof 1. It indicates that when $X$ and $Y$ are highly correlated (e.g. a clear image with a matching caption), $f(X,Y)$ is large; AdaIB thus prioritises preserving predictive information about $Z$ and $Y$.

**Theorem 2 (minimality at low relevance)** *Let $f : \mathcal{X} \times \mathcal{Y} \to (0,\infty)$ and $g : (0,\infty) \to (0,\infty)$ with $g$ nonincreasing. Assume $I(X;Y) < \infty$ and that $\lim_{u \to 0^+} \frac{g(u)}{u} = +\infty$. Then for every $\varepsilon > 0$ there exists $\eta > 0$ such that whenever $0 < f(X,Y) < \eta$, any maximizer $Z^*$ of $\mathcal{L}_{\text{AdaIB}}$ satisfies*

$$I(Z^*;X) \ \le \ \inf_Z I(Z;X) \ + \ \varepsilon.$$

*Thus, as $f \to 0^+$ (hence $g(f)/f \to \infty$), the objective is dominated by the compression term and the solution approaches a minimal-information representation.*

Theorem 2 is proved in Appendix B.2 Proof 2. It indicates that in the low-relevance regime, the AdaIB objective is dominated by the compression term, $-g(f)\,I(Z;X)$. Hence, any maximiser $Z^\star$ approaches a representation that minimises $I(Z;X)$ within the model class, avoiding overfitting to noise or mismatched pairs. This behaviour reflects the IB principle of minimality.

**Theorem 3 (adaptive sufficiency–minimality trade-off)** *Let $f : \mathcal{X} \times \mathcal{Y} \to (0,\infty)$ be the relevance score and $g : (0,\infty) \to (0,\infty)$ be positive and non-increasing. Assume $I(Z;X) < \infty$ and $I(Z;Y) < \infty$ for all admissible $Z$, and define the effective compression weight $\lambda(f) := g(f)/f$. Then $\lambda(f)$ is non-increasing in $f$ (see Lemma B.1), and the AdaIB objective factors as*

$$\mathcal{L}_{\text{AdaIB}} = f\Big[I(Z;Y) - \lambda(f)\,I(Z;X)\Big].$$

Consequently, as $f$ increases, $\lambda(f)$ decreases monotonically while the sufficiency term is scaled by $f$, implementing a sample-wise trade-off that shifts emphasis from compression to sufficiency. The limiting behaviours as $f \to 0^+$ and $f \to \infty$ are characterised by Theorems 1 and 2, respectively. The proof of Theorem 3 (Appendix B.2, Proof 4) confirms that between these extremes, the objective follows the adaptive trade-off principle.

In addition, AdaIB also satisfies a bounded leakage property, ensuring it never incentivises gratuitous dependence on $X$ at fixed predictive power. (see Appendix B.3).

At any fixed sufficiency level $I(Z;Y)$, AdaIB strictly prefers representations with smaller $I(Z;X)$, and even in the high-relevance limit, it never increases $I(Z;X)$ without also improving $I(Z;Y)$. Detailed arguments are deferred to Appendix B.2.

## 4.4 LEARNABLE RELEVANCE AND COMPRESSION FUNCTIONS

We now extend AdaIB by allowing both the relevance function $f$ and the compression function $g$ to be learned from data as independent functions $f_\theta$ and $g_\phi$. In Sections 4.1–4.3, we assumed $g$ was a nonincreasing function of $f$, enforcing a specific monotonic relationship.

**Definition 2 (decoupled learnable functions)** *We parameterise independent relevance and compression functions:*

$$f_\theta(X,Y) = \epsilon_f + \text{act}\big(h_\theta(X,Y)\big), \qquad g_\phi(X,Y) = \epsilon_g + \text{act}\big(u_\phi(X,Y)\big),$$

*where $\epsilon_f, \epsilon_g > 0$ ensure strict positivity, and $\text{act}(\cdot)$ is a nonnegative activation. The AdaIB objective becomes:*

$$\mathcal{L}_{\text{AdaIB}} = f_\theta(X,Y)\,I(Z;Y) \ - \ g_\phi(X,Y)\,I(Z;X).$$

Under Definition 2, we establish in Appendix B.4 that stationary optimal points still exist. Moreover, the properties developed in Sections 4.1–4.3 continue to hold after decoupling (see Appendix B.5). In particular, the adaptive sufficiency–minimality trade-off remains valid (Appendix B.6).

In practice, different degrees of flexibility can be considered, such as using a fixed $f$ with learnable $g_\phi(f)$, a learnable $f_\theta(X,Y)$ with fixed $g(f)$, or jointly learning both $f_\theta(X,Y)$ and $g_\phi(X,Y)$. These variants extend the applicability of AdaIB across heterogeneous settings.

# 5 EXPERIMENTS

## 5.1 DATASETS AND BASELINES

In this study, we adopt the experimental setup from M2IB (Wang et al., 2023) and NIB (Zhu et al., 2025), leveraging the pre-trained CLIP model with a Vision Transformer (ViT-B/32) (Radford et al., 2021) as the visual encoder. CLIP's ability to jointly align visual and textual modalities has shown remarkable performance across various multimodal tasks.

While prior work often focuses on relatively small datasets such as Flickr8k (Hodosh et al., 2013), which includes 8,000 images paired with natural language descriptions. Additionally, we aim to evaluate the model's generalisation ability as thoroughly as possible. To this end, we conduct experiments on larger and more diverse datasets, specifically Conceptual Captions 3M (CC3M) (Sharma et al., 2018) and LAION-400M (Schuhmann et al., 2021), both of which provide large-scale image-text pairs suitable for learning robust multimodal representations. CC3M consists of automatically generated image-text alignments from the web, offering a rich training signal for vision-language learning. LAION-400M further expands this scale with hundreds of millions of image-text pairs, enabling comprehensive evaluation of the model's generalisation capabilities across diverse domains. More details about the datasets can be found in Appendix A.1.

For baselines, we compare against several well-established attribution techniques to evaluate their effectiveness. The baseline methods include NIB (Zhu et al., 2025), M2IB (Wang et al., 2023), RISE (Petsiuk et al., 2018), Grad-CAM (Selvaraju et al., 2017), the method by (Chefer et al., 2021), Saliency Maps (Simonyan, 2013), MFABA (Zhu et al., 2024), and FastIG (Hesse et al., 2021).

## 5.2 EXPERIMENTAL SETTINGS

Following the approach of MI2B and NIB, we insert an information bottleneck into a specified layer of both the text and image encoders within the CLIP model for each "image-caption" pair. To train the bottleneck, we adopt the same procedure as the Per-Sample Bottleneck method from IBA (Schulz et al., 2020), repeating each sample 10 times to stabilise optimisation. We optimise by 10 steps using Adam with a learning rate of 1. To further enhance stability, we also apply gradient clipping during optimisation, specifically capping the global L2-norm of the gradients at 1.0. For the function $f(X, Y)$, we choose the L2 distance by default. The function $g(f(X, Y))$ is implemented as a learnable shallow MLP with a $1 \rightarrow 32 \rightarrow 1$ architecture and a ReLU activation function. We keep $f$ to be heuristically chosen and $g$ to be trainable, which we found to be optimal based on our experimental results. Detailed ablation studies concerning these choices for $f$ and $g$ are discussed in the Appendix E. All results are reported as the mean ± standard deviation over 5 independent runs to ensure statistical reliability. All experiments were performed on a single 4090 GPU. The detailed experimental setup can be found in Appendix A.2.

## 5.3 EVALUATION METRICS

Consistent with prior work (Wang et al., 2023; Zhu et al., 2025), we evaluate the quality of our generated attribution maps using a comprehensive suite of metrics from (Chattopadhay et al., 2018; Hooker et al., 2019).

**Confidence drop.** This metric quantifies the drop in model confidence when only the most salient features are preserved. A high-quality attribution method should identify features that are sufficient for the model's prediction; consequently, their preservation should lead to only a minimal drop in confidence. It is computed as: Drop $= \frac{1}{N} \sum_{i=1}^{N} \max(0, o_i - s_i)$ where $o_i$ and $s_i$ denote the original and post-masking image-text cosine similarities, respectively. Lower values are better.

**Confidence increase.** Conversely, this metric evaluates whether removing irrelevant features reduces noise and thereby *increases* the model's confidence. It is defined as the proportion of samples for which the confidence improves after masking: Incr. $= \frac{1}{N} \sum_{i=1}^{N} \mathbb{I}(o_i < s_i)$ where $\mathbb{I}(\cdot)$ is the indicator function. Higher values are better.

Table 1: Comparison of interpretability methods across multiple vision-language datasets. We report the mean and standard deviation (*mean ± std*) on both image and text modalities using two metrics: Drop (Confidence Drop) and Incr. (Confidence Increase)

| Dataset | Method | M2IB | Grad-CAM | Chefer et al. | MFABA | FastIG | NIB | AdaIB (Ours) |
|---------|--------|------|----------|---------------|-------|--------|-----|--------------|
| cc3M-I | Drop ↓ | **1.11** ± 0.10 | 8.07 ± 0.23 | 6.78 ± 0.11 | 2.38 ± 0.11 | 1.02 ± 0.07 | 1.03 ± 0.10 | **1.01** ± 0.08 |
| cc3M-I | Incr. ↑ | 37.20 ± 3.75 | 7.20 ± 1.52 | 11.00 ± 2.26 | 21.90 ± 2.63 | 32.90 ± 2.48 | 38.80 ± 3.63 | **40.70** ± 2.60 |
| cc3M-T | Drop ↓ | **0.90** ± 0.08 | 1.79 ± 0.14 | 1.02 ± 0.08 | 1.60 ± 0.18 | 1.42 ± 0.15 | 1.13 ± 0.10 | 1.07 ± 0.11 |
| cc3M-T | Incr. ↑ | 37.80 ± 3.91 | 33.90 ± 1.52 | 37.00 ± 3.55 | 27.30 ± 4.02 | 38.80 ± 4.19 | **40.50** ± 2.65 | 40.10 ± 2.07 |
| Flickr8k-I | Drop ↓ | 1.59 ± 0.07 | 9.19 ± 0.38 | 6.72 ± 0.21 | 2.52 ± 0.20 | 1.58 ± 0.13 | 1.49 ± 0.10 | **1.26** ± 0.10 |
| Flickr8k-I | Incr. ↑ | 27.50 ± 1.46 | 5.10 ± 2.16 | 8.30 ± 1.64 | 16.40 ± 1.08 | 18.90 ± 3.85 | 28.30 ± 2.44 | **29.80** ± 1.44 |
| Flickr8k-T | Drop ↓ | 1.43 ± 0.14 | 2.34 ± 0.11 | 1.49 ± 0.13 | 2.16 ± 0.22 | 2.07 ± 0.23 | 1.38 ± 0.19 | **1.18** ± 0.20 |
| Flickr8k-T | Incr. ↑ | 36.00 ± 1.77 | 30.90 ± 2.16 | 38.30 ± 3.29 | 25.60 ± 3.40 | 35.00 ± 4.00 | **42.40** ± 1.98 | 41.70 ± 3.98 |
| Laion400m-I | Drop ↓ | 1.57 ± 0.08 | 11.28 ± 0.31 | 9.83 ± 0.33 | 3.41 ± 0.27 | 1.52 ± 0.15 | 1.65 ± 0.10 | **1.26** ± 0.16 |
| Laion400m-I | Incr. ↑ | 30.52 ± 2.53 | 1.10 ± 1.15 | 3.51 ± 0.62 | 14.66 ± 2.22 | 28.82 ± 3.30 | 31.62 ± 2.70 | **33.81** ± 1.24 |
| Laion400m-T | Drop ↓ | 1.38 ± 0.14 | 2.53 ± 0.16 | 1.45 ± 0.19 | 2.16 ± 0.24 | 1.90 ± 0.20 | 1.50 ± 0.20 | **1.09** ± 0.23 |
| Laion400m-T | Incr. ↑ | 30.43 ± 3.68 | 28.31 ± 2.59 | 32.43 ± 3.41 | 22.09 ± 2.27 | **35.56** ± 4.41 | 35.55 ± 3.83 | 34.93 ± 3.68 |

Table 2: The quantitative comparison of our method against several baselines on the Refcoco dataset, using various metrics including pointing-game IoU and Drop/Incr.

| Metric | M2IB | Grad-CAM | Chefer et al. | MFABA | FastIG | NIB | AdaIB (Ours) |
|--------|------|----------|---------------|-------|--------|-----|--------------|
| vdrop↓ | 0.84 ± 0.06 | 1.98 ± 0.09 | 1.09 ± 0.03 | 1.59 ± 0.01 | 0.86 ± 0.01 | 0.72 ± 0.09 | **0.72** ± 0.05 |
| vincr↑ | 45.52 ± 1.69 | 46.38 ± 0.18 | 46.00 ± 0.64 | 29.90 ± 0.99 | 35.52 ± 0.04 | 48.16 ± 2.98 | **50.20** ± 1.92 |
| tdrop↓ | 1.02 ± 0.10 | 1.97 ± 0.06 | **1.00** ± 0.04 | 1.64 ± 0.03 | 1.49 ± 0.04 | 1.21 ± 0.05 | **1.00** ± 0.07 |
| tincr↑ | 41.16 ± 2.00 | 37.20 ± 1.27 | 40.50 ± 0.21 | 31.38 ± 0.25 | 37.40 ± 0.57 | 41.88 ± 1.38 | **44.12** ± 2.85 |
| mIoU↑ | 14.20 ± 0.57 | 8.89 ± 0.12 | 10.97 ± 0.01 | 4.90 ± 0.13 | 9.47 ± 0.10 | 11.96 ± 0.33 | **16.46** ± 0.32 |

**Pointing-game IoU.** To evaluate the spatial grounding performance of the attribution map, we employ the Pointing-Game Intersection over Union (IoU) on the RefCOCO dataset (Kazemzadeh et al., 2014). This metric quantifies the overlap between a binary mask, generated by thresholding the attribution map, and the ground-truth bounding box of the referenced object. Higher IoU is better. The threshold used to generate the binary mask from the attribution map is set to 0.5 by default.

**Remove and retrain (ROAR).** We also adapt the computationally efficient Remove and Retrain (ROAR) benchmark (Hooker et al., 2019). The purpose of this benchmark is to assess how critical the features identified by our method are to the model's predictions. This is achieved by removing the most salient features and measuring the subsequent drop in zero-shot image-text retrieval performance. The ROAR score is calculated by the formula $\frac{ACC_o - ACC_c}{Acc_o}$, where $ACC_o$ is the accuracy on the original data and $ACC_c$ is the accuracy on the data after feature removal. A higher score indicates a more effective attribution method. A detailed description of our implementation is available in Appendix F.

## 5.4 EXPERIMENTAL RESULTS

**Quantitative results:** Table 1, Table 2 and Table 3 present a comparison of attribution performance across multiple vision-language datasets using three different metrics. On the Refcoco dataset, AdaIB achieves the highest pointing-game mIoU of 16.46 ± 0.32, significantly outperforming all baselines. This result highlights its ability to generate attribution maps that are not only accurate but also spatially precise. Furthermore, on the ROAR metric, AdaIB consistently leads on diverse datasets like cc3M, Flickr8k, and Laion400m for both image-to-text (i2t) and text-to-image (t2i) retrieval tasks. For instance, on the Flickr8k dataset, AdaIB achieves the best scores for both i2t-oc (66.95 ± 2.23) and t2i-oc (71.85 ± 2.03). Additionally, AdaIB shows strong performance on the Drop and Increase metrics, securing the lowest Drop scores in multiple tasks (e.g. 1.01 ± 0.08 for cc3M-I) and the highest Increase scores (e.g. 29.80 ± 1.44 for Flickr8k-I). These comprehensive results confirm that AdaIB not only generates reliable and accurate attributions but also exhibits strong robustness and generalizability across various complex datasets and tasks. **Qualitative results:** The comparison of the attribution method visualisation results of different methods can be seen in the Appendix G.

Table 3: Quantitative results of the ROAR metric, which evaluates zero-shot performance on image-to-text (i2t) and text-to-image (t2i) retrieval tasks under different corruption settings. For *i2t-oc*, the image features are original (o) while the text features are corrupted (c). For *i2t-co*, the image is corrupted (c) and the text is original (o). The same logic applies to the t2i metrics. An upward arrow (↑) indicates that higher scores are better. The best result in each row is highlighted in **bold**.

| Dataset | Metric | M2IB | Grad-CAM | Chefer et al. | MFABA | FastIG | NIB | AdaIB (Ours) |
|---|---|---|---|---|---|---|---|---|
| cc3M-I | i2t-oc↑ | $48.96_{\pm 3.52}$ | $57.01_{\pm 1.80}$ | $44.68_{\pm 1.75}$ | $40.65_{\pm 0.85}$ | $47.10_{\pm 1.79}$ | $58.98_{\pm 2.10}$ | $\mathbf{60.86}_{\pm 2.21}$ |
| | i2t-co↑ | $68.20_{\pm 2.46}$ | $57.33_{\pm 1.65}$ | $34.01_{\pm 1.88}$ | $17.03_{\pm 1.47}$ | $52.62_{\pm 2.58}$ | $57.98_{\pm 1.24}$ | $\mathbf{68.97}_{\pm 3.54}$ |
| cc3M-T | t2i-oc↑ | $51.92_{\pm 3.30}$ | $62.73_{\pm 1.88}$ | $40.96_{\pm 1.93}$ | $46.40_{\pm 1.78}$ | $42.99_{\pm 1.34}$ | $\mathbf{63.53}_{\pm 1.96}$ | $63.27_{\pm 1.92}$ |
| | t2i-co↑ | $56.13_{\pm 0.73}$ | $47.73_{\pm 2.41}$ | $35.38_{\pm 1.96}$ | $10.20_{\pm 2.16}$ | $45.47_{\pm 2.33}$ | $48.62_{\pm 1.72}$ | $\mathbf{56.95}_{\pm 2.22}$ |
| Flickr8k-I | i2t-oc↑ | $64.31_{\pm 2.27}$ | $63.22_{\pm 3.16}$ | $47.15_{\pm 2.50}$ | $41.06_{\pm 1.68}$ | $42.80_{\pm 3.20}$ | $61.69_{\pm 3.24}$ | $\mathbf{66.95}_{\pm 2.21}$ |
| | i2t-co↑ | $51.43_{\pm 2.60}$ | $45.38_{\pm 1.24}$ | $29.62_{\pm 1.84}$ | $20.38_{\pm 1.31}$ | $40.80_{\pm 3.17}$ | $46.34_{\pm 1.72}$ | $\mathbf{52.16}_{\pm 2.03}$ |
| Flickr8k-T | t2i-oc↑ | $68.75_{\pm 1.78}$ | $65.21_{\pm 3.13}$ | $51.00_{\pm 1.54}$ | $44.75_{\pm 1.77}$ | $47.47_{\pm 3.40}$ | $64.34_{\pm 0.28}$ | $\mathbf{71.85}_{\pm 2.03}$ |
| | t2i-co↑ | $50.47_{\pm 2.15}$ | $43.02_{\pm 2.63}$ | $28.09_{\pm 2.80}$ | $22.20_{\pm 2.47}$ | $40.50_{\pm 2.50}$ | $43.96_{\pm 2.80}$ | $\mathbf{51.20}_{\pm 1.26}$ |
| Laion400m-I | i2t-oc↑ | $48.78_{\pm 3.81}$ | $46.53_{\pm 3.10}$ | $46.46_{\pm 1.67}$ | $38.85_{\pm 2.93}$ | $49.50_{\pm 1.02}$ | $52.01_{\pm 2.94}$ | $\mathbf{57.53}_{\pm 2.62}$ |
| | i2t-co↑ | $67.85_{\pm 1.44}$ | $42.89_{\pm 2.50}$ | $43.68_{\pm 1.51}$ | $13.14_{\pm 2.32}$ | $34.72_{\pm 2.47}$ | $57.34_{\pm 2.18}$ | $\mathbf{68.11}_{\pm 2.33}$ |
| Laion400m-T | t2i-oc↑ | $50.40_{\pm 4.17}$ | $47.95_{\pm 2.51}$ | $37.97_{\pm 1.50}$ | $31.04_{\pm 2.09}$ | $49.98_{\pm 1.65}$ | $53.81_{\pm 2.59}$ | $\mathbf{59.62}_{\pm 2.98}$ |
| | t2i-co↑ | $56.88_{\pm 0.90}$ | $43.14_{\pm 3.07}$ | $43.99_{\pm 3.42}$ | $10.03_{\pm 1.51}$ | $37.66_{\pm 3.89}$ | $48.50_{\pm 2.60}$ | $\mathbf{58.55}_{\pm 1.47}$ |

## 6 ANALYSIS OF THE ADAIB

**Misalignment image-caption analysis.** We discussed the performance of AdaIB on misalignment image-caption pairs in Appendix C. Our method maintains leading performance under artificially set misalignment of image and caption. While the other baseline models fail to distinguish the matched and mismatched image-caption pairs (Fig. 3 in Appendix).

**Dynamic $\beta$ of AdaIB.** In Appendix D, we study and demonstrate the changing relationship between $f$ and $g$ in AdaIB under different data samples. Interestingly, we found that for different samples under the similar $f$ will produce different $g$ values, indicating that AdaIB's adaptability is not solely based on the image-caption distance calculated by $f$, but can also be adjusted based on the intrinsic information of the data itself. This is consistent with the different optimal beta values for different samples shown in Fig. 2.

**Computational cost.** We report the computational cost and memory consumption in Table 4. The core of our method introduces only a shallow MLP with a single layer of additional parameters and feature distance calculation, resulting in no significant increase in computational load or memory consumption compared to the original CLIP model.

Table 4: Computational efficiency of AdaIB compared to baselines M2IB and NI

| Metric | M2IB | NIB | AdaIB (Ours) |
|---|---|---|---|
| FPS | 2.47 | 12.5 | 2.27 |
| Memory | 3.28GB | 2.32GB | 3.28GB |

## 7 LIMITATIONS AND CONCLUSION

We propose AdaIB, a powerful attribution framework that relaxes the common assumption of strict semantic alignment in multimodal explanations. While many existing methods can overfit or produce misleading attributions when faced with misaligned inputs, AdaIB takes a different approach. Grounded in the Information Bottleneck (IB) principle, it dynamically balances compression and prediction terms based on the relationship between modalities. Without enforcing strict pairwise alignment, AdaIB adapts automatically to various input conditions, delivering more reliable and interpretable explanations. This provides a more robust and dependable approach to multimodal interpretability in real-world settings.

However, despite its strengths, our framework has limitations. We focus on attribution under semantic misalignment in large-scale web-scraped datasets (e.g. CC3M, LAION-400M). While AdaIB addresses such cases effectively, we do not examine its performance on subtler mismatches, such as sarcasm, puns, or metaphors. These non-literal relationships remain a significant challenge for attribution, and future work will explore extending AdaIB to handle them.

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

## THE USE OF LARGE LANGUAGE MODELS (LLMS)

In the preparation of this manuscript, we employed large language models (LLMs) to assist with improving the clarity and readability of the text. Specifically, LLMs were used to refine grammar, enhance fluency, and polish the overall presentation of ideas, while all conceptual contributions, analyses, and experimental results remain the sole work of the authors.

## REPRODUCIBILITY STATEMENT

We have made every effort to ensure the reproducibility of our work. All training protocols, model architectures, hyperparameters, and evaluation settings are described in detail in the main text and appendix. Upon acceptance of this paper, we will release our implementation and experimental code in a public GitHub repository to further facilitate reproducibility and future research.

## A    IMPLEMENTATION DETAILS.

### A.1    DATASETS

In our study, we evaluate the model's performance and generalisation capabilities on large-scale, diverse datasets. Below are the detailed descriptions of the primary datasets used for training and evaluation.

**Flickr8k**    Flickr8k (Hodosh et al., 2013) is a classic and widely-used benchmark dataset in the field of image captioning. It consists of 8,000 images collected from the Flickr website. The defining characteristic of this dataset is its high-quality, human-generated annotations. Each image is paired with five independent, descriptive sentences written by human annotators. Compared to large-scale, web-crawled datasets, Flickr8k is significantly smaller but features clean, reliable captions. In this work, it serves as a baseline to contrast with the large-scale, noisy data environments where we aim to test our model's generalisation capabilities.

**Conceptual Captions 3M (CC3M)**    The Conceptual Captions 3M dataset (Sharma et al., 2018) is a large-scale collection of approximately 3.3 million image-URL and caption pairs, automatically sourced from alt-text attributes of images on the web. Unlike curated datasets like Flickr8k, the captions in CC3M are not human-generated annotations but are naturally occurring descriptions. While this results in a higher level of noise and less descriptive detail, the sheer scale and diversity of the data provide a rich training signal for learning robust vision-language alignments.

**LAION-400M Subset**    LAION-400M (Schuhmann et al., 2021) is an open-source, massive-scale dataset containing approximately 400 million image-text pairs scraped from the web. The pairs were filtered using CLIP to ensure a baseline level of semantic alignment between the image and its corresponding text.

Due to the immense size of the full dataset and computational constraints, our experiments were conducted on a specific subset of LAION-400M. Specifically, we utilised the initial data shards numbered from 00000 to 01241. Each shard contains approximately 25,000 samples. Our selection of 1,242 shards (from 0 to 1,241 inclusive) results in a subset of approximately 31 million image-text pairs that constitutes roughly 7.8% of the full 400-million-pair dataset. This selected portion is substantial enough to ensure robust model training while remaining computationally manageable compared to the previous datasets used in the baselines.

**RefCOCOg (UMD)**    The RefCOCOg dataset was collected through an interactive game setting. For our experiments, we use the version from the Hugging Face Hub under the identifier lmms-lab/RefCOCOg. This specific dataset contains approximately 7.57k image-text pairs, each with a corresponding segmentation mask for the target object. The referring expressions in this dataset are significantly longer, more complex, and use more conversational language, posing a greater challenge for language understanding.

## A.2 Experimental Settings

Our experimental setup is adapted from the protocol of M2IB (Zhu et al., 2025), as the setup for NIB was not detailed in their work. We expand on M2IB (Wang et al., 2023)'s approach to create a more robust evaluation framework. To ensure statistical reliability, all experiments were independently repeated 5 times, using a different random seed for each run (42, 43, 44, 45, and 46). The final results are consistently reported as the mean and standard deviation of these runs.

Our sampling strategy varies by dataset to balance thoroughness and computational cost. For the smaller, task-specific datasets RefCOCOg and Flickr8k, we used the entirety of the data in each of the 5 runs to conduct a comprehensive evaluation. For the large-scale CC3M and LAION-400M datasets, we randomly sampled 2,000 pairs for each run to efficiently assess the model's generalisation performance on diverse data. In M2IB (Zhu et al., 2025), only 500 pairs were sampled per run.

## B  Theoretical Proof

### B.1  Variational Derivation Steps

**Step 1: Variational lower bound for $I(Z; Y)$.**  For the sufficiency term, we introduce a variational distribution $q(y|z)$ to approximate the true posterior $p(y|z)$.

$$I(Z;Y) = \mathbb{E}_{p(z,y)}\left[\log \frac{p(y|z)}{p(y)}\right] = \mathbb{E}_{p(z,y)}[\log q(y|z)] + H(Y) + \mathbb{E}_{p(z)}\big[\mathrm{KL}\big(p(y|z)\,\|\,q(y|z)\big)\big]. \quad (5)$$

This identity follows by adding and subtracting $\log q(y|z)$ inside the expectation and using $\mathbb{E}_{p(z,y)}[\log p(y)] = \mathbb{E}_{p(y)}[\log p(y)] = -H(Y)$ and $\mathbb{E}_{p(z,y)}[\log p(y|z) - \log q(y|z)] = \mathbb{E}_{p(z)}[\mathrm{KL}(p(y|z)\,\|\,q(y|z))]$. By the non-negativity of KL divergence, dropping the last term yields the lower bound.

$$I(Z;Y) \ \geq \ \mathbb{E}_{p(z,y)}[\log q(y|z)] + H(Y). \quad (6)$$

**Step 2: Variational upper bound for $I(Z; X)$.**  For the compression term, we approximate the intractable marginal $p(z)$ with a variational distribution $r(z)$ (independent of $x$). Then

$$\mathbb{E}_{p(x)}\big[\mathrm{KL}(p(z|x)\,\|\,r(z))\big] = \mathbb{E}_{p(x,z)}\left[\log \frac{p(z|x)}{r(z)}\right] \quad (7)$$

$$= \mathbb{E}_{p(x,z)}\left[\log \frac{p(z|x)}{p(z)}\right] + \mathbb{E}_{p(z)}\left[\log \frac{p(z)}{r(z)}\right] \quad (8)$$

$$= I(Z;X) + \mathrm{KL}\big(p(z)\,\|\,r(z)\big). \quad (9)$$

By non-negativity of KL,

$$I(Z;X) \ \leq \ \mathbb{E}_{p(x)}\big[\mathrm{KL}(p(z|x)\,\|\,r(z))\big], \quad (10)$$

with equality when $r(z) = p(z)$.

**Step 3: Variational lower bound for AdaIB.**  Substituting the bounds from Steps 1–2 into the AdaIB objective (Def. 1), and taking the outer expectation over $(X, Y)$, we obtain the lower bound.

$$\mathcal{L}_{\mathrm{AdaIB}}^{\mathrm{var}} = \mathbb{E}_{p(x,y)}\Big[f(X,Y)\,\mathbb{E}_{p(z|x)}[\log q(y|z)] \ - \ g\big(f(X,Y)\big)\,\mathrm{KL}\big(p(z|x)\,\|\,r(z)\big)\Big]. \quad (11)$$

The additive term $H(Y)\,\mathbb{E}_{p(x,y)}[f(X,Y)]$ does not depend on $q, p, r$ and is omitted for optimisation over these parameters.

**Step 4: Empirical approximation.**  Given $N$ i.i.d. samples $\{(x_i, y_i)\}_{i=1}^{N}$, Eq. equation 11 is estimated by Monte Carlo:

$$\hat{\mathcal{L}} = \frac{1}{N}\sum_{i=1}^{N}\Big[f(x_i,y_i)\log q(y_i|z_i) \ - \ g\big(f(x_i,y_i)\big)\,\mathrm{KL}\big(p(z|x_i)\,\|\,r(z)\big)\Big], \quad z_i \sim p(z|x_i), \quad (12)$$

where $z_i$ is drawn via the reparameterization trick to obtain low-variance gradients.

**Step 5: Practical instantiations of $f$ and $g$.** To complete the specification of AdaIB, we propose stable and practical choices for the relevance function $f$ and the compression mapping $g$. We consider two types of $f$ that ensure larger values indicate stronger relevance:

**(i) Similarity-based:**

$$s(x,y) = \frac{1+\cos(x,y)}{2} \in [0,1], \quad f(x,y) = \mathrm{softplus}(\tau s(x,y)) + \epsilon_f, \quad \tau > 0, \ \epsilon_f > 0.$$

Here $\cos$ is computed on $\ell_2$-normalized features.

**(ii) Inverse-distance-based:**

$$d(x,y) = \|x-y\|_p, \quad p \in \{1,2\}, \quad f(x,y) = \frac{1}{d(x,y)+\epsilon_f}, \quad \epsilon_f > 0.$$

For the compression mapping, we adopt

$$g(f) = \frac{1}{f+\epsilon_g}, \quad \epsilon_g > 0,$$

which ensures stable optimisation by preventing excessively large weights when $f$ is small. Substituting these choices into the empirical objective equation 12 yields the final, concrete objective function:

$$\hat{\mathcal{L}} = \frac{1}{N}\sum_{i=1}^{N}\left[ f(x_i,y_i)\,\log q(y_i|z_i) - \frac{1}{f(x_i,y_i)+\epsilon_g}\,\mathrm{KL}\big(p(z|x_i)\,\|\,r(z)\big)\right]. \tag{13}$$

### B.2 SUFFICIENCY AND MINIMALITY PRINCIPALS

**Proof 1 (Sufficiency at high relevance)** *Rewrite the objective as*

$$\mathcal{L}_{\mathrm{AdaIB}} = f(X,Y)\left[ I(Z;Y) - \frac{g(f(X,Y))}{f(X,Y)}\,I(Z;X)\right].$$

*Let $t := f(X,Y)$ and define $\varepsilon(t) := \frac{g(t)}{t}$. Since $g$ is positive and non-increasing on $(0,\infty)$, the limit $c := \lim_{t\to\infty} g(t) \in [0,\infty)$ exists; hence*

$$\varepsilon(t) = \frac{g(t)}{t} \ \leq\ \frac{\max\{c,g(1)\}}{t} \ \longrightarrow\ 0.$$

*Because $I(Z;X) < \infty$, we have $\varepsilon(t)\,I(Z;X) \to 0$, and thus the bracketed term converges to $I(Z;Y)$. Multiplying by $f(X,Y) = t$ yields*

$$\mathcal{L}_{\mathrm{AdaIB}} \sim f(X,Y)\cdot I(Z;Y),$$

*where "$\sim$" denotes asymptotic equivalence (ratio $\to 1$) as $t \to \infty$.*

**Proof 2 (Minimality at low relevance)** *Rewrite $\mathcal{L}_{\mathrm{AdaIB}}(Z) = -\,g(t)\big(I_X(Z) - \delta(t)I_Y(Z)\big)$ with $\delta(t) := \frac{f(t)}{g(t)}$. Let $\varepsilon > 0$ and pick $Z_{\min}$ such that $I_X(Z_{\min}) \leq \inf_Z I_X(Z) + \varepsilon/2$. Optimality gives $\mathcal{L}(Z^*) \geq \mathcal{L}(Z_{\min})$, hence*

$$I_X(Z^*) - \delta(t)I_Y(Z^*) \ \leq\ I_X(Z_{\min}) - \delta(t)I_Y(Z_{\min}),$$

*so*

$$I_X(Z^*) - I_X(Z_{\min}) \ \leq\ \delta(t)\big(I_Y(Z^*) - I_Y(Z_{\min})\big) \ \leq\ \delta(t)\big|I_Y(Z^*) - I_Y(Z_{\min})\big|.$$

*By the data processing inequality $I(Z;Y) \leq I(X;Y)$, we have $\big|I_Y(Z^*) - I_Y(Z_{\min})\big| \leq 2\,I(X;Y) =: 2M_Y$. Therefore*

$$I_X(Z^*) - I_X(Z_{\min}) \ \leq\ 2\,\delta(t)\,M_Y.$$

*Since $\delta(t) = f(t)/g(t) \to 0$ as $f(t) \to 0^+$, choose $t_\varepsilon$ so that $2\,\delta(t)M_Y \leq \varepsilon/2$ for $t < t_\varepsilon$. Then*

$$I_X(Z^*) \ \leq\ I_X(Z_{\min}) + \varepsilon/2 \ \leq\ \inf_Z I_X(Z) + \varepsilon,$$

**Lemma B.1 (Monotonicity of the compression)** *Define the effective compression weight $\lambda(f) := \frac{g(f)}{f}$. If $g : (0, \infty) \to (0, \infty)$ is positive and non-increasing, then $\lambda(f)$ is non-increasing on $(0, \infty)$.*

**Proof 3 (Monotonicity of the compression)** *Take $0 < f_1 < f_2$. Since $g$ is non-increasing and positive, $g(f_1) \geq g(f_2) > 0$. Then*

$$\frac{\lambda(f_2)}{\lambda(f_1)} = \frac{g(f_2)/f_2}{g(f_1)/f_1} = \Big(\frac{g(f_2)}{g(f_1)}\Big)\Big(\frac{f_1}{f_2}\Big) \leq 1 \cdot \frac{f_1}{f_2} < 1,$$

*hence $\lambda(f_2) \leq \lambda(f_1)$.*

**Proof 4 (Adaptive sufficiency–minimality trade-off)** *Algebraically factor $f$ from Definition 1 to obtain the displayed form. Monotonicity of $\lambda$ follows from Lemma B.1. When $f$ grows, the bracketed objective reduces the penalty coefficient on $I(Z; X)$ while scaling $I(Z; Y)$ by a larger $f$, hence shifting the balance toward sufficiency. The extremes follow from Theorems 1 and 2.*

### B.3 Bounded Leakage

**Proposition 2 (No Gratuitous Leakage)** *Let $\mathcal{Z}_c := \{Z : I(Z; Y) = c\}$ be the set of representations achieving the same predictive information level $c \geq 0$. For any fixed relevance score $f > 0$ (and thus $g(f) > 0$ by Definition 1), and for any $Z_1, Z_2 \in \mathcal{Z}_c$ satisfying $I(Z_1; X) > I(Z_2; X)$, it holds that*

$$\mathcal{L}_{\mathrm{AdaIB}}(Z_1) < \mathcal{L}_{\mathrm{AdaIB}}(Z_2).$$

*Therefore, at a fixed sufficiency level, AdaIB always prefers the representation with smaller $I(Z; X)$.*

**Proof 5 (No Gratuitous Leakage)** *For fixed $f > 0$,*

$$\mathcal{L}_{\mathrm{AdaIB}}(Z) = f \cdot I(Z; Y) - g(f) \cdot I(Z; X).$$

*Given $I(Z_1; Y) = I(Z_2; Y) = c$,*

$$\mathcal{L}_{\mathrm{AdaIB}}(Z_1) - \mathcal{L}_{\mathrm{AdaIB}}(Z_2) = - g(f) \big(I(Z_1; X) - I(Z_2; X)\big) < 0,$$

*since $g(f) > 0$ and $I(Z_1; X) > I(Z_2; X)$.*

**Corollary B.1 (Bounded leakage at high relevance)** *Let $Z^\star$ be an optimizer of $\mathcal{L}_{\mathrm{AdaIB}}$ within a given model class, and write $c^\star := I(Z^\star; Y)$. Then for any $f > 0$ there is no $Z' \in \mathcal{Z}_{c^\star}$ with $I(Z'; X) > I(Z^\star; X)$. In particular, along any sequence $f \to \infty$ (with $g(f) > 0$), AdaIB never incentivizes increasing $I(Z; X)$ without improving $I(Z; Y)$; among equal-$I(Z; Y)$ solutions, the optimizer attains minimal $I(Z; X)$.*

**Proof 6 (Bounded leakage at high relevance)** *By Proposition 2, for any fixed $f > 0$ and any $Z_1, Z_2 \in \mathcal{Z}_{c^\star}$, if $I(Z_1; X) > I(Z_2; X)$ then $\mathcal{L}_{\mathrm{AdaIB}}(Z_1) < \mathcal{L}_{\mathrm{AdaIB}}(Z_2)$ because $g(f) > 0$. Hence an optimizer $Z^\star$ at that $f$ must minimize $I(Z; X)$ within $\mathcal{Z}_{c^\star}$. This argument holds for every $f$; therefore, it also holds along any sequence with $f \to \infty$ (regardless of whether $g(f)$ is bounded), which proves the claim.*

**Synthesis.** AdaIB provides a principled adaptive mechanism for balancing sufficiency and minimality:

- *Prioritizes sufficiency* when relevance is high (Theorem 1).
- *Enforces minimality* when relevance is low (Theorem 2).
- *Protects against overfitting* even at high relevance by preferring the smallest $I(Z; X)$ among equally sufficient solutions (Corollary B.1).

This adaptive behaviour, controlled by the relevance function $f(X, Y)$, lets AdaIB adjust its learning strategy to data quality, which is particularly useful in multimodal settings with heterogeneous and noisy pairs.

## B.4 Existence of Stationary Points

**Theorem 4 (Existence of Stationary Points)** *Let $\hat{\mathcal{L}}(w)$ denote the empirical AdaIB objective in equation 12, with decoupled $f_\theta, g_\phi$, and let $w = (\theta, \phi, \psi)$ collect all trainable parameters. Assume:*

(A1) ***Positivity and boundedness.*** *$f_\theta(x, y) \in [\varepsilon_f, M_f]$ and $g_\phi(x, y) \in [\varepsilon_g, M_g]$ for all $(x, y)$, with $\varepsilon_f, \varepsilon_g > 0$.*

(A2) ***Smoothness on compact domain.*** *$\hat{\mathcal{L}}$ is continuously differentiable on a nonempty compact convex set $\Omega \subset \mathbb{R}^d$, with $\nabla\hat{\mathcal{L}}$ bounded on $\Omega$.*

(A3) ***Projected gradient dynamics.*** *Training uses $T(w) = \Pi_\Omega(w - \eta\nabla\hat{\mathcal{L}}(w))$ for some $\eta > 0$, where $\Pi_\Omega$ is Euclidean projection onto $\Omega$.*

*Then $T : \Omega \to \Omega$ is continuous and admits a fixed point $w^* \in \Omega$ by Brouwer's fixed-point theorem (Brouwer, 1911). Consequently, $w^*$ satisfies the stationarity condition $0 \in \nabla\hat{\mathcal{L}}(w^*) + N_\Omega(w^*)$, where $N_\Omega$ is the normal cone. If $w^*$ lies in the relative interior of $\Omega$, then $\nabla\hat{\mathcal{L}}(w^*) = 0$.*

**Proof 7 (Existence of Stationary Points)** *We verify the conditions for Brouwer's fixed-point theorem, which states that every continuous map from a nonempty, compact, convex set to itself has a fixed point.*

*By assumption (A3), the training algorithm uses the projected gradient map $T(w) = \Pi_\Omega(w - \eta\nabla\hat{\mathcal{L}}(w))$. Since $\Pi_\Omega$ projects onto $\Omega$, we have $T(w) \in \Omega$ for any $w \in \Omega$, establishing that $T$ maps $\Omega$ to itself.*

*The continuity of $T$ follows from the continuous differentiability of $\hat{\mathcal{L}}$ (by A2) and the fact that the projection operator $\Pi_\Omega$ is nonexpansive (hence continuous) on the convex set $\Omega$. Thus, $T$ is a composition of continuous maps.*

*Since $\Omega$ is nonempty, compact, and convex, and $T : \Omega \to \Omega$ is continuous, Brouwer's theorem guarantees the existence of a fixed point $w^* \in \Omega$ satisfying $T(w^*) = w^*$.*

*The fixed point condition $w^* = \Pi_\Omega(w^* - \eta\nabla\hat{\mathcal{L}}(w^*))$ implies, by the projection theorem, that $-\eta\nabla\hat{\mathcal{L}}(w^*) \in N_\Omega(w^*)$, where $N_\Omega(w^*)$ is the normal cone to $\Omega$ at $w^*$. This is equivalent to the stationarity condition $0 \in \nabla\hat{\mathcal{L}}(w^*) + N_\Omega(w^*)$. If $w^*$ lies in the relative interior of $\Omega$, then $N_\Omega(w^*) = \{0\}$ and thus $\nabla\hat{\mathcal{L}}(w^*) = 0$.*

## B.5 Decouple Properties

Despite decoupling, the following properties still hold:

(i) **Pointwise reparameterization** (Prop. 1): with $\lambda(x, y) := g_\phi(x, y)/f_\theta(x, y) > 0$, $\mathcal{L}_{\text{AdaIB}} = f_\theta(x, y)\,[\,I(Z; Y) - \lambda(x, y)\,I(Z; X)\,]$.

(ii) **Sample-wise reweighting**: letting $f_i = f_\theta(x_i, y_i)$, $g_i = g_\phi(x_i, y_i)$, $w_i^{(y)} = f_i / \sum_j f_j$, $w_i^{(x)} = g_i / \sum_j g_j$, and $\bar{\beta}_{\text{eff}} = \sum_i g_i / \sum_i f_i$, we obtain the same weighted IB form.

(iii) **No gratuitous leakage** (Prop. 2): for fixed $(x, y)$, if $I(Z_1; Y) = I(Z_2; Y)$ and $I(Z_1; X) > I(Z_2; X)$, then $\mathcal{L}_{\text{AdaIB}}(Z_1) < \mathcal{L}_{\text{AdaIB}}(Z_2)$. This uses only $g_\phi(x, y) > 0$ and does not require any functional dependence between $g$ and $f$.

## B.6 Decouple Sufficiency and Minimality

**Sufficiency and Minimality Balance** Recall the decoupled objective $\mathcal{L}_{\text{AdaIB}} = f_\theta(x, y)\,[\,I(Z; Y) - \lambda(x, y)\,I(Z; X)\,]$ with $\lambda(x, y) := g_\phi(x, y)/f_\theta(x, y) > 0$. We do *not* assume any global monotonic relation between $g_\phi$ and $f_\theta$. The extreme-regime behaviours and their quantitative, per-sample approximations remain valid under mild ratio conditions.

Even without a global monotone constraint on $g$, the two extreme behaviours are retained under local ratio conditions. Let $K_X, K_Y < \infty$ be uniform bounds on $I(Z;X)$ and $I(Z;Y)$ within the model class, and define $\lambda(x,y) := g_\phi(x,y)/f_\theta(x,y) > 0$.

**Theorem 5 ($\varepsilon$-sufficiency)** *Let $\lambda(x,y) := g_\phi(x,y)/f_\theta(x,y) > 0$ and $K_X < \infty$ bound $I(Z;X)$ on the model class. If a sample $(x,y)$ satisfies $\lambda(x,y) \le \varepsilon$ for some $\varepsilon > 0$, then*

$$\left| \mathcal{L}_{\mathrm{AdaIB}} - f_\theta(x,y)\, I(Z;Y) \right| \;\le\; \varepsilon\, f_\theta(x,y)\, K_X.$$

*Hence, the objective behaves as $f_\theta\, I(Z;Y)$ up to $O(\varepsilon)$; sufficiency dominates.*

**Theorem 6 ($\eta$-minimality)** *Let $K_Y < \infty$ bound $I(Z;Y)$ on the model class. If a sample $(x,y)$ satisfies $\frac{f_\theta(x,y)}{g_\phi(x,y)} \le \eta$ for some $\eta > 0$ (equivalently $\lambda(x,y) \ge 1/\eta$), then*

$$\left| \mathcal{L}_{\mathrm{AdaIB}} + g_\phi(x,y)\, I(Z;X) \right| \;\le\; \eta\, g_\phi(x,y)\, K_Y.$$

*Hence maximizing $\mathcal{L}_{\mathrm{AdaIB}}$ is approximately equivalent to minimizing $I(Z;X)$ up to $O(\eta)$; minimality dominates.*

On a minibatch, let $\Pi_{\mathrm{hi}} := \{i : \lambda(x_i, y_i) \le \varepsilon\}$ and $\Pi_{\mathrm{lo}} := \{i : f_\theta(x_i, y_i)/g_\phi(x_i, y_i) \le \eta\}$. Then $\hat{\mathcal{L}}$ can be viewed as the sum of a sufficiency-dominant average over $\Pi_{\mathrm{hi}}$ (error $O(\varepsilon)$) and a minimality-dominant average over $\Pi_{\mathrm{lo}}$ (error $O(\eta)$). Thus, the sufficiency–minimality balance is preserved in a quantitative, per-sample manner.

Here is the operational behaviour under Definition 2.

- **High relevance.** Along sequences where $f_\theta(x,y) \to \infty$ and $\lambda(x,y) \to 0$ (e.g. $g_\phi$ is bounded or grows sublinearly relative to $f_\theta$), we have $\mathcal{L}_{\mathrm{AdaIB}} \sim f_\theta(x,y)\, I(Z;Y)$; compression vanishes and sufficiency dominates.

- **Low relevance.** Along sequences where $f_\theta(x,y) \to 0^+$ and $\lambda(x,y) \to \infty$ (equivalently $f_\theta/g_\phi \to 0$), we have $\mathcal{L}_{\mathrm{AdaIB}} = -g_\phi(x,y)\, I(Z;X) + o\big(g_\phi(x,y)\big)$; maximizing the objective asymptotically minimizes $I(Z;X)$; compression dominates.

## C  FURTHER ANALYSIS ON MODEL PERFORMANCE UNDER MISALIGNMENT

To provide a more comprehensive analysis of our model's robustness against image-text misalignment, we conducted two supplementary experiments.

First, to probe this capability in a more controlled setting, we constructed a synthetic dataset with deliberately mismatched pairs. We sampled 2,000 image-text pairs from CC3M, Flickr8k, and Laion400m, respectively, and created a balanced set of correct (original) and incorrect (randomly swapped captions) pairs. We then evaluated whether the models could

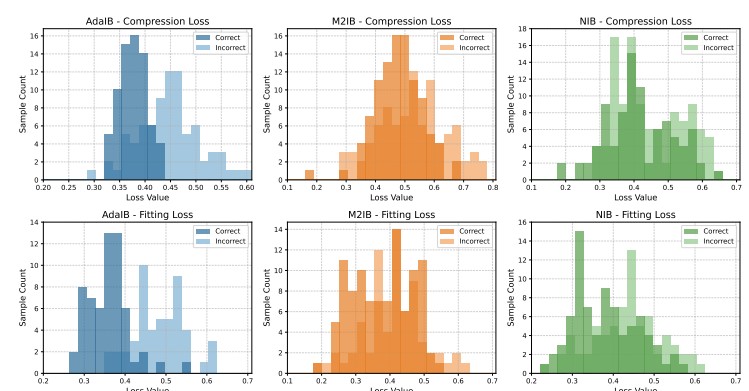

Figure 3: Distributions of compression and fitting losses under matched pairs vs. mismatched pairs. AdaIB enables separation, while M2IB and NIB show significant overlap.

distinguish between these two categories based on their final loss values, using the Area Under the Curve (AUC) as a ranking metric. The results presented in Fig. 3, demonstrate the distributions of

Table 5: Performance comparison of M2IB, NIB, and Ours across noisy⋆, borderline⋆, and clean⋆ CC3M, Flickr8k, and Laion400m datasets. The asterisk (⋆) indicates that the datasets have been artificially partitioned into three distinct groups—noisy, borderline, and clean—based on the degree of image-caption alignment. This categorisation was achieved by manually partitioning the data according to the image-text similarity scores obtained from the CLIP model.

| Dataset | Metric | M2IB | NIB | Ours |
|---|---|---|---|---|
| CC3M (Noisy) | vdrop | $0.52 \pm 0.05$ | $0.47 \pm 0.05$ | $\mathbf{0.43 \pm 0.06}$ |
| CC3M (Noisy) | vincr | $57.57 \pm 4.83$ | $59.68 \pm 4.12$ | $\mathbf{60.01 \pm 4.75}$ |
| CC3M (Noisy) | tdrop | $1.01 \pm 0.08$ | $0.86 \pm 0.13$ | $\mathbf{0.85 \pm 0.12}$ |
| CC3M (Noisy) | tincr | $39.92 \pm 3.28$ | $44.21 \pm 4.47$ | $\mathbf{45.21 \pm 5.21}$ |
| CC3M (Borderline) | vdrop | $1.12 \pm 0.08$ | $1.02 \pm 0.08$ | $\mathbf{0.94 \pm 0.06}$ |
| CC3M (Borderline) | vincr | $33.28 \pm 2.61$ | $36.17 \pm 4.34$ | $\mathbf{37.47 \pm 2.42}$ |
| CC3M (Borderline) | tdrop | $1.01 \pm 0.14$ | $1.10 \pm 0.11$ | $\mathbf{0.98 \pm 0.07}$ |
| CC3M (Borderline) | tincr | $37.31 \pm 2.01$ | $\mathbf{40.53 \pm 2.22}$ | $39.23 \pm 0.61$ |
| CC3M (Clean) | vdrop | $1.92 \pm 0.44$ | $2.03 \pm 0.45$ | $\mathbf{1.54 \pm 0.34}$ |
| CC3M (Clean) | vincr | $22.73 \pm 1.33$ | $23.94 \pm 4.29$ | $\mathbf{29.49 \pm 3.15}$ |
| CC3M (Clean) | tdrop | $\mathbf{1.33 \pm 0.21}$ | $1.37 \pm 0.15$ | $1.35 \pm 0.23$ |
| CC3M (Clean) | tincr | $28.66 \pm 1.43$ | $29.49 \pm 2.84$ | $\mathbf{31.81 \pm 5.11}$ |
| Flickr8k (Noisy) | vdrop | $\mathbf{0.47 \pm 0.02}$ | $0.59 \pm 0.28$ | $0.48 \pm 0.14$ |
| Flickr8k (Noisy) | vincr | $47.50 \pm 3.54$ | $47.50 \pm 3.54$ | $\mathbf{53.57 \pm 5.05}$ |
| Flickr8k (Noisy) | tdrop | $1.30 \pm 0.27$ | $1.39 \pm 0.08$ | $\mathbf{1.25 \pm 0.26}$ |
| Flickr8k (Noisy) | tincr | $31.79 \pm 4.55$ | $25.71 \pm 6.06$ | $\mathbf{32.50 \pm 3.54}$ |
| Flickr8k (Borderline) | vdrop | $1.39 \pm 0.07$ | $1.18 \pm 0.06$ | $\mathbf{1.13 \pm 0.06}$ |
| Flickr8k (Borderline) | vincr | $30.36 \pm 4.88$ | $32.63 \pm 2.08$ | $\mathbf{34.35 \pm 4.91}$ |
| Flickr8k (Borderline) | tdrop | $1.39 \pm 0.10$ | $1.38 \pm 0.09$ | $\mathbf{1.31 \pm 0.14}$ |
| Flickr8k (Borderline) | tincr | $38.32 \pm 0.91$ | $41.16 \pm 1.48$ | $\mathbf{41.61 \pm 4.36}$ |
| Flickr8k (Clean) | vdrop | $2.30 \pm 0.09$ | $2.21 \pm 0.31$ | $\mathbf{1.85 \pm 0.08}$ |
| Flickr8k (Clean) | vincr | $10.20 \pm 1.06$ | $\mathbf{12.56 \pm 2.28}$ | $12.45 \pm 2.01$ |
| Flickr8k (Clean) | tdrop | $1.60 \pm 0.08$ | $1.49 \pm 0.00$ | $\mathbf{1.42 \pm 0.03}$ |
| Flickr8k (Clean) | tincr | $35.72 \pm 4.06$ | $\mathbf{41.73 \pm 2.24}$ | $37.96 \pm 3.10$ |
| Laion400m (Noisy) | vdrop | $1.21 \pm 0.62$ | $1.44 \pm 0.15$ | $\mathbf{1.15 \pm 0.50}$ |
| Laion400m (Noisy) | vincr | $24.36 \pm 12.69$ | $24.04 \pm 1.36$ | $\mathbf{32.05 \pm 1.81}$ |
| Laion400m (Noisy) | tdrop | $1.39 \pm 0.02$ | $1.60 \pm 0.27$ | $\mathbf{1.20 \pm 0.61}$ |
| Laion400m (Noisy) | tincr | $24.04 \pm 1.36$ | $43.27 \pm 25.84$ | $\mathbf{43.59 \pm 14.50}$ |
| Laion400m (Borderline) | vdrop | $1.56 \pm 0.15$ | $1.48 \pm 0.17$ | $\mathbf{1.29 \pm 0.15}$ |
| Laion400m (Borderline) | vincr | $29.21 \pm 2.66$ | $30.76 \pm 1.77$ | $\mathbf{32.77 \pm 0.65}$ |
| Laion400m (Borderline) | tdrop | $1.35 \pm 0.03$ | $1.38 \pm 0.06$ | $\mathbf{1.24 \pm 0.02}$ |
| Laion400m (Borderline) | tincr | $31.54 \pm 0.19$ | $\mathbf{38.02 \pm 2.89}$ | $34.94 \pm 1.98$ |
| Laion400m (Clean) | vdrop | $1.82 \pm 0.31$ | $1.84 \pm 0.21$ | $\mathbf{1.55 \pm 0.35}$ |
| Laion400m (Clean) | vincr | $25.79 \pm 0.46$ | $30.12 \pm 1.80$ | $\mathbf{31.43 \pm 2.11}$ |
| Laion400m (Clean) | tdrop | $\mathbf{1.26 \pm 0.03}$ | $1.45 \pm 0.19$ | $1.23 \pm 0.13$ |
| Laion400m (Clean) | tincr | $29.24 \pm 0.81$ | $\mathbf{33.63 \pm 1.63}$ | $33.02 \pm 0.14$ |

compression and fitting losses. Our model exhibits a clear separation between the loss distributions for matched and mismatched pairs. In stark contrast, the distributions for M2IB and NIB show significant overlap, suggesting their struggle to differentiate between well-matched and mismatched inputs.

Then, we analysed performance on data with varying degrees of naturally occurring misalignment. Leveraging CLIP similarity scores as a proxy for alignment quality, we partitioned the CC3M, Flickr8k, and Laion400m datasets into three distinct subsets: 'Noisy' (low similarity), 'Borderline' (medium similarity), and 'Clean' (high similarity). The detailed performance breakdown, presented in Table 5, shows that our model consistently outperforms the M2IB and NIB baselines across these partitions. Notably, our method's superiority is evident even on the most challenging "noisy" subset (in this case, the completely misaligned ones), highlighting its resilience to the imperfections inherent in web-crawled data.

## D ANALYSIS OF THE ADAPTIVE MECHANISM'S BEHAVIOR

The Fig. 4 visualises the distribution of matched and mismatched image-caption pairs and their corresponding $f$ and $g$ values. The $f$ value, representing the L2 distance, demonstrates a clear discriminative capability. Matched image-caption pairs are predominantly associated with higher $f$ values, indicating that this metric effectively captures the overall alignment between the two modalities. In contrast, mismatched pairs are more concentrated in the lower range of $f$ values.

A key finding, however, is the significant variability in the corresponding $g$ values for both matched and mismatched pairs. The $g$ value corresponding to the same $f$ value is not fixed. This suggests that the **AdaIB optimisation process does not rely solely on the initial $f$ value. Instead, the**

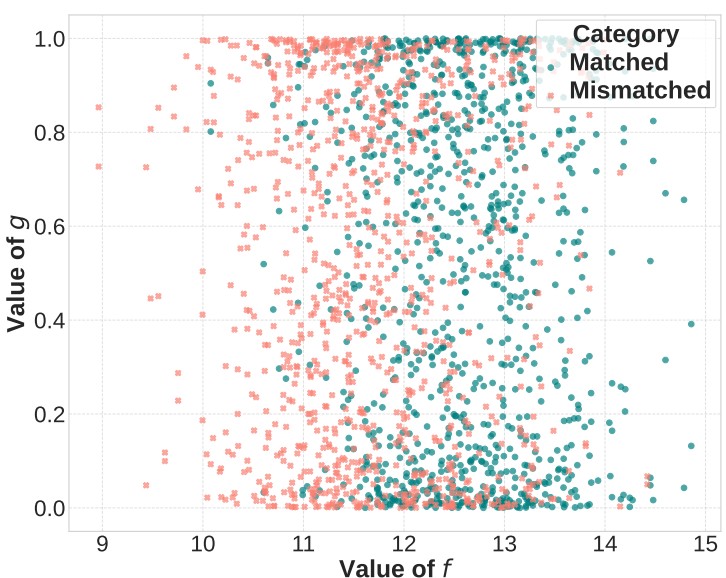

Figure 4: The visualisation of distribution of $f$ and $g$ values from the Laion400m dataset. Matched pairs represent the original, aligned image-caption pairs, while Mismatched pairs are misaligned image-caption pairs created by randomly shuffling the original image-caption pairings.

**framework appears to learn more nuanced, context-dependent characteristics of the data**. For instance, a high $f$ value might still lead to high compression (high $g$) if the image contains irrelevant background elements, while a low $f$ value could result in low compression if the model cannot find a meaningful signal and defaults to a near-uniform distribution. This behaviour demonstrates that the optimal compression weight $g$ is a function of deeper data properties beyond initial similarity metrics, highlighting the adaptive and non-linear nature of the AdaIB framework.

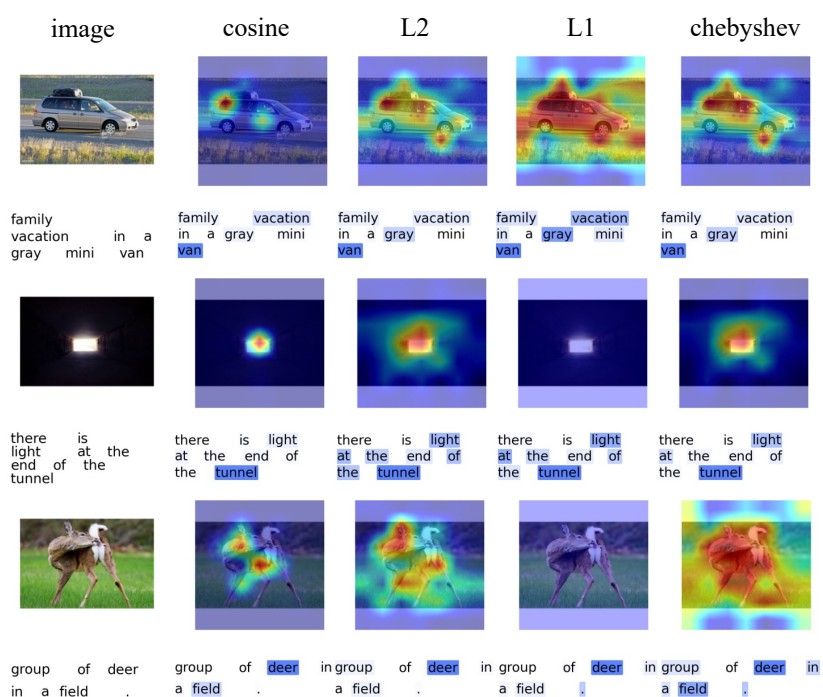

Figure 5: A qualitative comparison of attribution maps generated using various measurements for the function $f(X, Y)$. The results for L1, L2, Chebyshev, and Cosine functions illustrate that the metric selection critically influences the model's visual attribution, causing it to attend to different spatial regions for the same textual concept.

# E   MORE ABLATION STUDY

## E.1   THE ABLATION STUDY ON $f(X, Y)$

To investigate the impact of the similarity function $f(X, Y)$ on the model's performance and its visual attribution, an ablation study was conducted. The study systematically evaluated four different metric choices for f: L1 distance, L2 distance, Chebyshev distance, and Cosine similarity. The quantitative results are presented in Table 6.

Table 6: Ablation study on the choice of function $F$ and different metrics across datasets and ROAR protocols.

(a) Performance on different datasets.

| Metric | | CC3M | Flickr8k | Laion400m |
|---|---|---|---|---|
| $F$=L1 | vdrop ↓ | **0.75±0.10** | 0.78±0.04 | 0.92±0.03 |
| | vincr ↑ | 22.50±3.54 | 22.50±4.95 | 31.66±0.49 |
| | tdrop ↓ | 1.67±0.26 | 2.44±0.47 | 1.89±0.07 |
| | tincr ↑ | 31.50±3.54 | 30.00±12.73 | 27.14±1.61 |
| $F$=L2 | vdrop ↓ | 0.79±0.15 | **0.56±0.04** | **0.64±0.06** |
| | vincr ↑ | **53.00±4.24** | 60.00±2.83 | **55.76±6.00** |
| | tdrop ↓ | **1.20±0.08** | **1.61±0.10** | 1.29±0.22 |
| | tincr ↑ | **36.00±2.83** | 37.50±0.71 | 28.63±3.35 |
| $F$=chebyshev | vdrop ↓ | 0.85±0.14 | **0.64±0.09** | 0.78±0.29 |
| | vincr ↑ | 47.00±1.41 | 54.00±4.24 | 53.74±7.44 |
| | tdrop ↓ | 1.35±0.05 | 1.76±0.30 | 1.27±0.20 |
| | tincr ↑ | 30.00±1.41 | 31.00±0.00 | 25.64±3.74 |
| $F$=cosine | vdrop ↓ | 2.21±0.12 | 1.15±0.01 | 2.96±0.08 |
| | vincr ↑ | 42.50±3.54 | **61.00±1.41** | 45.73±0.39 |
| | tdrop ↓ | 1.25±0.43 | 1.69±0.01 | **1.25±0.13** |
| | tincr ↑ | 29.50±2.12 | 33.50±2.12 | 23.12±1.59 |

(b) Performance on different ROAR protocols.

| Metric | | CC3M | Flickr8k | Laion400m |
|---|---|---|---|---|
| $F$=L1 | i2t-oc ↑ | 45.53±1.95 | 46.00±1.59 | 44.06±3.89 |
| | t2i-oc ↑ | 46.45±0.42 | 48.75±2.60 | 44.85±6.55 |
| | i2t-co ↑ | 31.18±9.17 | 17.60±0.50 | 31.14±5.61 |
| | t2i-co ↑ | 27.78±8.47 | 19.23±2.75 | 22.45±1.00 |
| $F$=L2 | i2t-oc ↑ | 44.82±6.45 | **82.40±0.50** | **60.74±1.36** |
| | t2i-oc ↑ | 47.72±5.04 | **85.56±1.21** | 59.35±0.92 |
| | i2t-co ↑ | **57.55±2.41** | 38.40±5.53 | 52.64±6.80 |
| | t2i-co ↑ | **46.75±6.05** | 42.90±3.72 | **41.13±4.36** |
| $F$=chebyshev | i2t-oc ↑ | 41.15±0.58 | 67.75±3.34 | 52.69±0.80 |
| | t2i-oc ↑ | 43.22±0.52 | 70.48±0.15 | 52.42±1.19 |
| | i2t-co ↑ | 58.72±7.74 | **40.36±6.31** | **53.19±4.51** |
| | t2i-co ↑ | 45.77±7.79 | 40.93±2.01 | 37.90±5.95 |
| $F$=cosine | i2t-oc ↑ | **53.72±4.34** | 73.96±0.56 | 49.43±3.81 |
| | t2i-oc ↑ | **55.47±3.14** | 76.56±3.46 | 46.49±2.73 |
| | i2t-co ↑ | 52.52±0.80 | 36.21±4.06 | 45.14±2.35 |
| | t2i-co ↑ | 44.50±4.16 | 37.31±2.14 | 34.23±0.78 |

Fig. 5 provides a qualitative comparison of attribution maps generated using the different similarity functions. It is evident that the selection of $f(X, Y)$ influences the model's visual attention, guiding it to focus on different spatial regions of the image that correspond to the textual query.

For the first example in Fig. 5, the L2 and Cosine similarity functions generate attribution maps that are more precisely focused on the vehicle itself, aligning well with the key object in the text. In contrast, the L1 and Chebyshev metrics produce more diffuse heatmaps, with the Chebyshev metric highlighting the vehicle's edges and surroundings more broadly. For the third instance in Fig 5, the L2 and Cosine metrics again demonstrate a stronger ability to localise the subject ("deer"), whereas the L1 and Chebyshev metrics result in more scattered attention, with Chebyshev's attribution being particularly dispersed across the background and foreground.

Overall, the qualitative results suggest that the L2 and Cosine similarity functions tend to produce more semantically focused and cleaner attribution maps, which more accurately highlight the most relevant objects described in the text. The quantitative results from the Table 6 further confirm this trend, with the L2-based model demonstrating superior performance under various datasets. We attribute that L2 distance takes into account the "strength" information of the feature (vector modulus), it can make more accurate judgments than cosine similarity, which only looks at the "direction" of the feature, thus achieving a consistent advantage in multimodal attribution.

### E.2 THE ABLATION STUDY ON $g(f(X, Y))$

A comprehensive analysis of the updated ablation study, presented in Table 7, was conducted to evaluate six different model architectures by varying their depth, width, and activation functions.

The results unequivocally identify Model 2, a shallow architecture with a single hidden layer ($1 \rightarrow 32 \rightarrow 1$) and a ReLU activation function, as the superior configuration. This model demonstrated the most robust performance, achieving the highest scores on four of the eight metrics (VIncr, TIncr, ROAR-t2t-oc, and ROAR-t2i-oc) and was the joint-best performer on a fifth metric (VDrop).

A detailed breakdown of the findings is as follows:

Table 7: Comprehensive results of the ablation study on model architectures, split into two parts. (a) details the model configurations. (b) shows all corresponding performance metrics.

(a) Model architectures and configurations.

| ID | Architecture | Activation | Purpose / Description |
|----|--------------|------------|------------------------|
| 1 | $1 \rightarrow 1$ | None | Depth Ablation (Linear Model) |
| 2 | $1 \rightarrow 32 \rightarrow 1$ | ReLU | Depth Ablation (Shallower) |
| 3 | $1 \rightarrow 8 \rightarrow 8 \rightarrow 1$ | ReLU | Width Ablation (Narrower) |
| 4 | $1 \rightarrow 64 \rightarrow 64 \rightarrow 1$ | ReLU | Width Variation (Wider) |
| 5 | $1 \rightarrow 32 \rightarrow 32 \rightarrow 1$ | None | Activation Ablation (Removed) |
| 6 | $1 \rightarrow 32 \rightarrow 32 \rightarrow 1$ | Tanh | Activation Ablation (Replaced) |

(b) Performance metrics for each model configuration on Flickr8k dataset. For Drop metrics, lower is better. For all other metrics, higher is better. Best results are highlighted in **bold**.

| ID | VDrop | VIncr | TDrop | TIncr | ROAR-i2t-oc | ROAR-t2i-oc | ROAR-i2t-co | ROAR-t2i-co |
|----|-------|-------|-------|-------|-------------|-------------|-------------|-------------|
| 1 | $0.64 \pm 0.09$ | $54.00 \pm 4.24$ | $\mathbf{1.76 \pm 0.30}$ | $31.00 \pm 0.00$ | $67.75 \pm 3.34$ | $70.48 \pm 0.15$ | $40.36 \pm 6.31$ | $40.93 \pm 2.01$ |
| 2 | $\mathbf{0.56 \pm 0.09}$ | $\mathbf{57.00 \pm 4.24}$ | $1.81 \pm 0.03$ | $\mathbf{35.50 \pm 0.71}$ | $\mathbf{73.12 \pm 5.50}$ | $\mathbf{76.45 \pm 3.36}$ | $37.62 \pm 1.19$ | $40.43 \pm 3.93$ |
| 3 | $0.58 \pm 0.04$ | $53.00 \pm 4.24$ | $1.96 \pm 0.13$ | $28.50 \pm 0.71$ | $64.74 \pm 2.72$ | $65.72 \pm 3.09$ | $39.41 \pm 0.47$ | $38.02 \pm 3.85$ |
| 4 | $\mathbf{0.56 \pm 0.06}$ | $55.00 \pm 1.41$ | $1.90 \pm 0.06$ | $30.50 \pm 0.71$ | $67.24 \pm 0.81$ | $70.48 \pm 0.15$ | $38.64 \pm 3.88$ | $\mathbf{44.60 \pm 1.52}$ |
| 5 | $0.58 \pm 0.04$ | $54.50 \pm 2.12$ | $\mathbf{1.76 \pm 0.09}$ | $33.50 \pm 0.71$ | $67.62 \pm 6.78$ | $71.63 \pm 3.52$ | $40.36 \pm 6.31$ | $39.87 \pm 6.47$ |
| 6 | $0.57 \pm 0.03$ | $55.00 \pm 1.41$ | $1.85 \pm 0.03$ | $30.50 \pm 0.71$ | $67.75 \pm 3.34$ | $69.86 \pm 1.03$ | $38.77 \pm 0.44$ | $39.78 \pm 1.36$ |

**Impact of Depth and Non-linearity** : The study highlights the distinct advantage of a shallow, non-linear architecture. Both the simple linear model (ID 1) and the deeper model without an

activation function (ID 5) performed poorly on most metrics, confirming that non-linearity is essential for the task. Crucially, the shallow Model 2 significantly outperformed all deeper architectures (ID 3, 4, and 6), suggesting that increasing network depth beyond a single hidden layer is counterproductive.

**Impact of Network Width** : Network width was also a significant factor. The wider architecture (ID 4, with 64 neurons) was a strong contender, achieving the top result for the ROAR-t2i-co metric and a joint-best score for VDrop. However, its overall performance did not surpass that of the more streamlined, shallower Model 2. In contrast, the narrower model (ID 3, with 8 neurons) proved to be ineffective, showing mediocre performance across all metrics.

**Impact of Activation Function** : The choice of activation function was a key determinant of performance. As noted, the absence of non-linearity (ID 5) was detrimental. In comparing non-linear functions, the ReLU activation used in the top-performing Model 2 proved to be far more effective than the Tanh activation used in Model 6, which yielded unremarkable results.

The ablation study demonstrates that a shallow network with a single hidden layer of 32 neurons, activated by the ReLU function, provides the optimal architecture among the configurations tested. This structure strikes the most effective balance, outperforming models that are deeper, wider, or that utilise an alternative (Tanh) or no activation function.

## F    VISUALISATION OF ROAR

We adapt the Remove and Retrain (ROAR) benchmark (Hooker et al., 2019) to create a more computationally efficient evaluation. Instead of the costly process of retraining the model from scratch, we leverage the powerful zero-shot capabilities of our CLIP-based architecture. The methodology follows the spirit of ROAR: we first identify and remove the

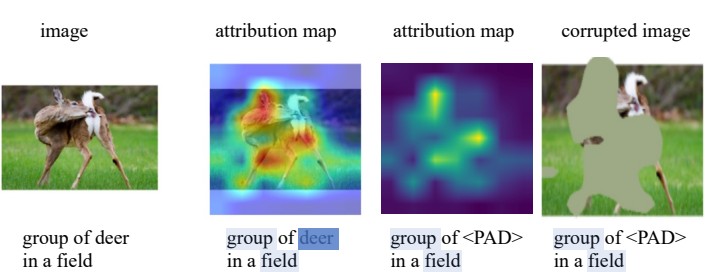

Figure 6: The visualisation process of ROAR.

most salient features to create a "corrupted" dataset. We then quantify the importance of these removed features by measuring the degradation in zero-shot image-text retrieval performance. The score is calculated by $\frac{ACC_o - ACC_c}{Acc_o}$. In our setting, however, $Acc_o$ represents the baseline zero-shot retrieval accuracy on the original data, whilst $Acc_c$ is the accuracy on the corrupted data. An effective attribution method should identify features critical to the model's performance; their removal will therefore cause a substantial drop in $Acc_c$, yielding a score close to 1. A higher score is better. Following the protocol of M2IB (Wang et al., 2023), we corrupt the inputs based on attribution score percentiles. For images, we identify pixels with scores above the 75th percentile and replace their values with the image's mean channel values. For text, given its sparse nature, tokens with scores exceeding the 90th percentile are replaced with the designated CLIP padding token (ID 49407). An example can be found in Fig. 6.

## G    MORE VISUALISATION OF ATTRIBUTION MAP

In this section, we provide the attribution maps generated by different methods, providing a visual comparison of their ability to localise salient objects and actions within an image. Each row focuses on a distinct scene. In comparison to established methods, our proposed AdaIB consistently yields sharper and more precise attribution maps. The visualisations demonstrate AdaIB's superior capacity to focus on salient regions—such as the soccer player and ball, the dog's limbs, and the duck's body—while in Fig. 7, effectively suppressing irrelevant background noise. This enhanced precision contrasts with the more diffuse or scattered activations often observed in the outputs of other models.

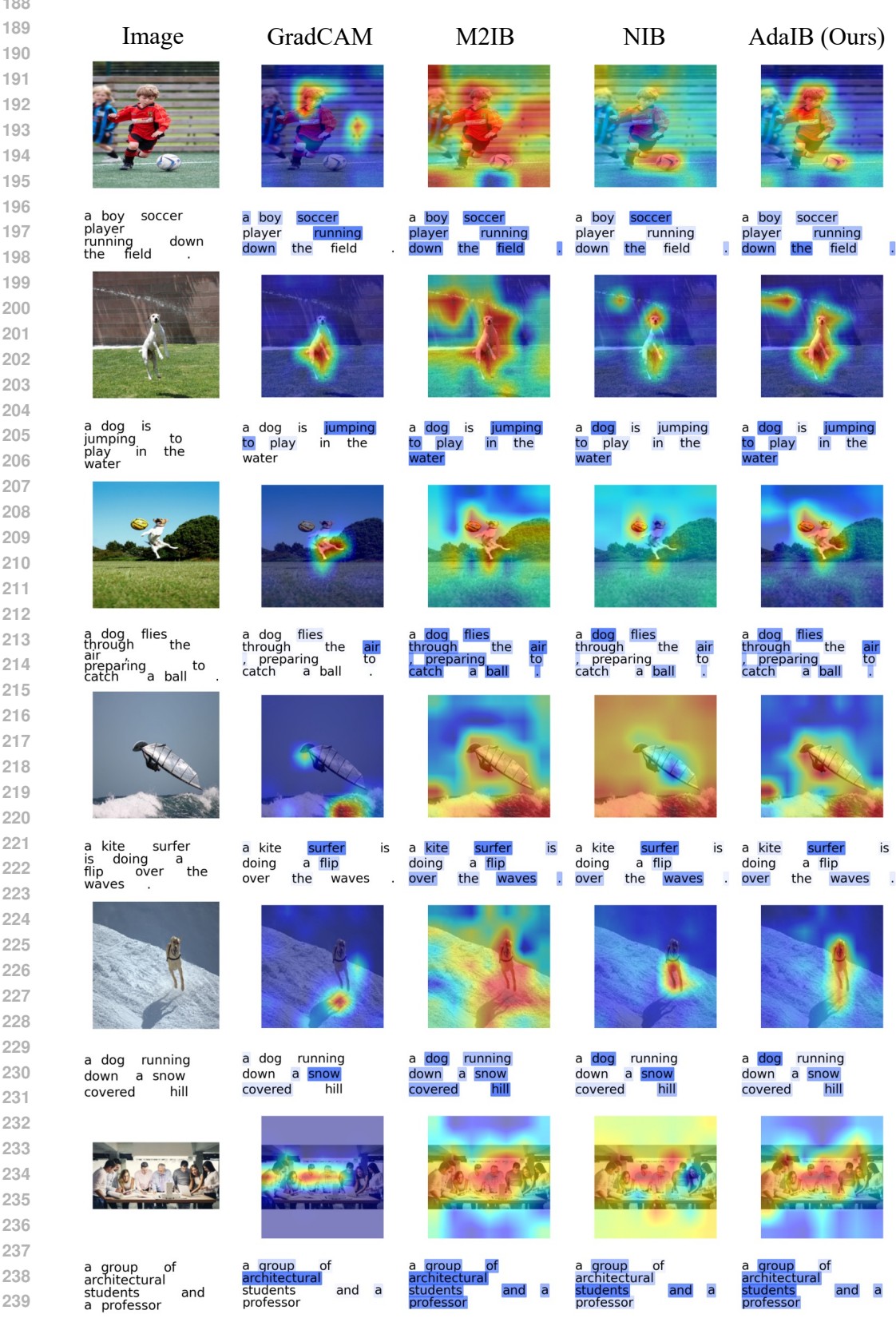

Figure 7: The visualisation of example attribution maps for image and text inputs.

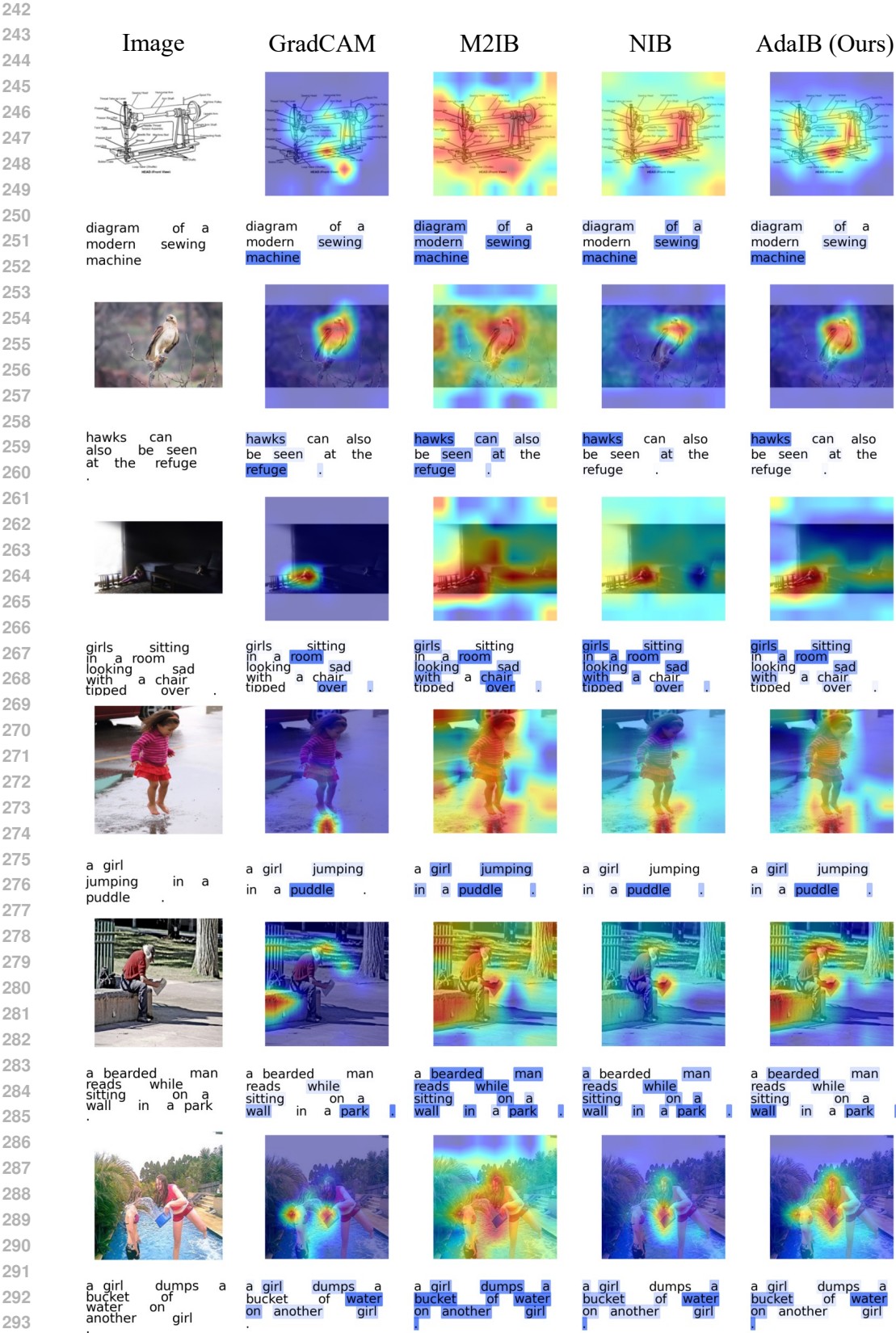

Figure 8: The visualisation of more example attribution maps for image and text inputs.

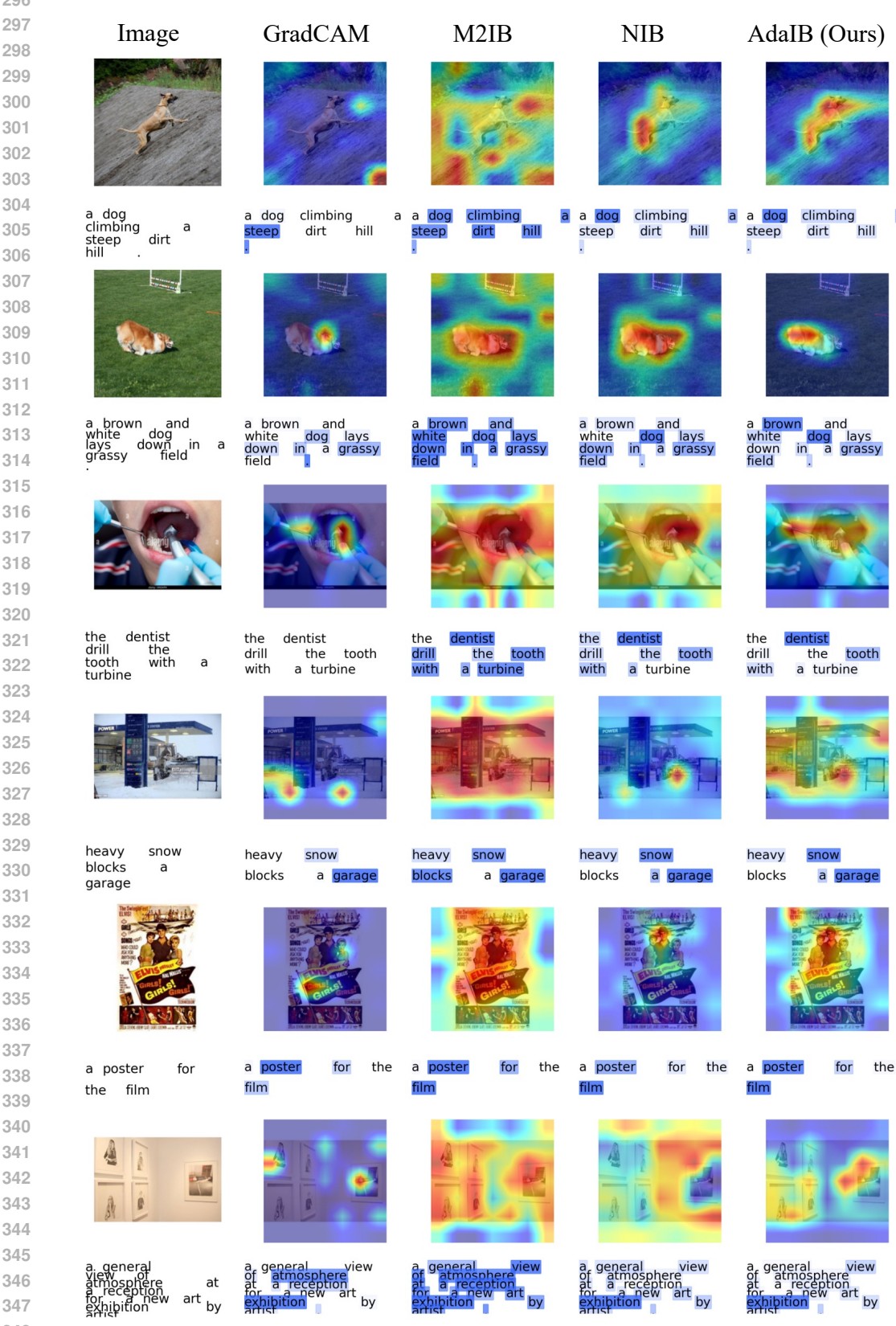

Figure 9: The visualisation of more example attribution maps for image and text inputs.

