# OpenReview forum: "Evil in the Pairing Assumption: Multimodal Attribution via Adaptive Information Bottleneck"
_ICLR.cc/2026/Conference — ICLR 2026 Conference Withdrawn Submission_

### Official Review · Reviewer_NkdQ · 2025-10-31

**Soundness:** 2
**Presentation:** 3
**Contribution:** 2
**Rating:** 2
**Confidence:** 4

**Summary:**

This paper proposes an Adaptive Information Bottleneck (AdaIB), which reformulates the classic IB objective to use sample-dependent weights, $f(X,Y)$ and $g(f)$, for the sufficiency and compression terms, respectively. The authors provide a theoretical analysis in Section 4.3 and Appendix B to ground this new objective, presenting theorems on its asymptotic behavior (sufficiency and minimality) and a proposition on "bounded leakage".

**Strengths:**

1. The paper is commendable for attempting to provide a principled, formal justification for its adaptive objective, rather than just presenting it as an empirical heuristic.
2. Proposition 1 clearly situates the classic IB as a special case of AdaIB, which is helpful for understanding the proposed generalization.
3. The variational derivation of the objective in Appendix B.1 appears to be standard and correct.

**Weaknesses:**

1. The main theoretical results (Theorems 1 and 2) are purely asymptotic, analyzing the behavior only as $f \to \infty$ or $f \to 0$. This provides little to no theoretical insight into the method's behavior in the "Moderate Fit" regime (as shown in Figure 2), which is presumably the most common and important case.
2. There appears to be a significant disconnect between the theory and the implementation.
    (i).  The primary theory (Sec 4.3, e.g., Theorem 3 assumes $g$ is a non-increasing *function of* $f$.
    (ii).  The actual implementation (Sec 5.2) *decouples* these: $f$ is a fixed L2 distance, and $g$ is an independently parameterized MLP.
    (iii).  While Appendices B.5 and B.6 *mention* this decoupled case, the main theoretical justifications presented in the paper (Sec 4.3) are based on the non-decoupled version. It is unclear if the guarantees from Sec 4.3, which rely on the $g(f)$ relationship, apply to the implemented model.
3.  Proposition 2 ("No Gratuitous Leakage") and its proof seem to be a trivial consequence of the objective function's definition. It merely states that for a fixed sufficiency $I(Z;Y)$, the term $-g(f) \cdot I(Z;X)$ will, by definition, prefer a smaller $I(Z;X)$ (since $g(f)>0$). This does not feel like a strong guarantee that information is not "leaking," especially in the high-$f$ regime where $g(f)$ (and thus the penalty) might be learned to be very small.

**Questions:**

1.  Do the authors have any theoretical guarantees for the non-asymptotic case (i.e., for $f$ in a moderate range $[a, b]$), which is not covered by Theorems 1 and 2?
2.  Regarding the theory-practice disconnect: Do the theoretical properties from Sec 4.3 (like the adaptive trade-off in Thm 3) hold for the *decoupled* implementation where $g$ is a separately learned MLP? Was any monotonicity of $g$ with respect to $f$ enforced during training?
3.  Can the authors provide an *empirical* validation of the "bounded leakage" claim (Prop 2)? For example, could a decoder trained to reconstruct $X$ from $Z$ demonstrate that, even for high-$f$ samples, $Z$ does not contain significantly more information about $X$ than necessary?

---

### Official Review · Reviewer_EUcm · 2025-10-31

**Soundness:** 3
**Presentation:** 3
**Contribution:** 3
**Rating:** 4
**Confidence:** 4

**Summary:**

This paper presents AdaIB, an adaptive extension of Information Bottleneck (IB) attribution methods. The core methodological contribution is to replace the fixed IB trade-off parameter ($\beta$) with a dynamic, sample-specific mechanism. This mechanism consists of two parts: a heuristic relevance function $f(X,Y)$ (instantiated as L2 distance) and a learnable compression function $g(f(X,Y))$ (instantiated as a shallow MLP). The authors provide extensive ablation studies to justify these design choices and demonstrate that this adaptive approach outperforms fixed-weight baselines.

**Strengths:**

1. The motivation for an adaptive trade-off is intuitive and well-illustrated by Figure 2. The idea that different samples require different compression levels is a clear and sensible improvement over a one-size-fits-all $\beta$.
2. The proposed $f$ and $g$ functions are a significant improvement over the more naive heuristics (e.g., $g=1/f$) discussed in prior work.
    * Designing $g$ as a learnable MLP is a strong choice, as it allows the model to learn complex compression strategies directly from the data.
    * The ablation studies in Appendix E (Tables 6 and 7) are comprehensive and provide strong empirical backing for the final design choices (L2 for $f$, and a $1 \to 32 \to 1$ ReLU MLP for $g$).
3. Figure 4 (Appendix D) is particularly insightful. It shows that even for samples with a similar $f$ value (L2 distance), the learned $g$ (compression weight) can vary wildly. This strongly supports the authors' claim that $g$ is capturing "nuanced, context-dependent characteristics" beyond what the simple L2 distance heuristic can provide.

**Weaknesses:**

1. While effective, the core idea is an extension of M2IB and NIB. The contribution of an "adaptive weight" is a solid, but arguably incremental, step rather than a major conceptual leap.
2. The authors motivate the need for a learnable, adaptive approach, yet they settle on a "half-fixed, half-learnable" design. $f(X,Y)$ is a fixed, hand-crafted heuristic (L2 distance), while $g$ is a flexible, learnable function. This seems inconsistent. Why not also learn the relevance function $f_{\theta}(X,Y)$, as the authors themselves propose in Definition 2?
3. The analysis of Figure 4 stops short. The paper claims $g$ learns "deeper data properties" but never defines what these might be. Is $g$ learning to compress more based on text length? Image entropy? Object count? Without this, the claim remains vague.

**Questions:**

1.  The authors propose a fully learnable $f_{\theta}(X,Y)$ in Definition 2. Did they experiment with this? Why was the final design choice a fixed L2 heuristic for $f$ but a learnable MLP for $g$?
2.  Following up on Figure 4: Can the authors provide a more concrete analysis of what the learnable $g$ function is actually learning? For example, can they show any correlation between the learned $g$ value and other data properties (e.g., text length, number of objects in the image, etc.)?
3.  How sensitive is the model to the specific architecture of $g$? Table 7 shows Model 2 ($1 \to 32 \to 1$) is best, but Model 4 ($1 \to 64 \to 64 \to 1$) also performs well. Is this choice stable across datasets, or must the $g$-network be carefully re-tuned for each new task?

---

### Official Review · Reviewer_DL8r · 2025-11-01

**Soundness:** 3
**Presentation:** 3
**Contribution:** 2
**Rating:** 2
**Confidence:** 3

**Summary:**

The paper argues that multimodal attribution methods (e.g., M2IB/NIB) implicitly assume well-aligned image–text pairs and can overfit under mismatch. It proposes AdaIB, which replaces the fixed IB coefficient with sample-adaptive weights: a relevance score f(X,Y) scales the sufficiency term I(Z;Y) and a compression weight g(f) (or a decoupled gϕ(X,Y)) scales I(Z;X). The authors derive a variational training objective, prove limiting properties (sufficiency/minimality), and report improved quantitative/qualitative attribution across CLIP-based benchmarks (CC3M, Flickr8k, LAION-400M, RefCOCOg).

**Strengths:**

1. The paper formalizes when fixed-β IB can under/overfit and proves that AdaIB recovers classical IB as a special case, with limiting behavior toward sufficiency (large 𝑓) and minimality (small 𝑓).
2. The derivation yields a practical per-sample loss with KL regularization against a variational prior r(z).
3. Results cover multiple datasets and metrics (Drop/Increase, ROAR, pointing-game IoU); tables show consistent gains over M2IB/NIB and gradient/perturbation baselines.

**Weaknesses:**

1. Some typos (with exact locations).
- “Following the approach of MI2B and NIB” → M2IB.
- “COCOA(Lin et al., 2022)” (missing space) → COCOA (Lin et al., 2022).
- Table-4 caption: “compared to baselines M2IB and NI” → NIB.
- Radford et al. (2021) venue string: “PmLR” → PMLR.
2. The text defines large f(X,Y) as high relevance (emphasizing sufficiency), but the experiments state “for f we choose the L2 distance by default,” where larger distance means lower relevance. This inverts the intended semantics unless an explicit inversion/monotone mapping is applied. The paper should either use similarity / inverse-distance (as later proposed) or clarify the transformation used in §5.2.
3. Theory in §4.1–4.3 assumes g is non-increasing in f, but §4.4 later decouples fθ(X,Y) and gϕ(X,Y) (both learnable). Without restoring a monotonicity constraint or proving new conditions, several theorems no longer directly apply. The paper claims properties “continue to hold,” but this needs a precise statement of assumptions and a proof sketch in the main text.
4. Eq. (13) proposes 𝑔(𝑓)=1/(𝑓+𝜖𝑔), which indeed yields strong compression when 𝑓→0. But combined with the distance-as-𝑓 choice in §5.2, this makes compression weaker for larger distances, contradicting the intended “compress more when relevance is low.” The paper should reconcile Eq. (13) with the actual 𝑓 used.
5. All results appear to use CLIP ViT-B/32; adding other backbones (e.g., RN50, ViT-L) would test robustness.
6. Beyond the main experiments (e.g., the ~31M-pair LAION subset), the paper includes a supplementary diagnostic that samples 2,000 pairs per dataset (CC3M/Flickr8k/LAION) to form balanced matched vs. mismatched sets for separability analysis (Fig. 3). Please make this distinction explicit in the experimental setup and consider reporting how conclusions scale from the 2,000-pair diagnostic to the full-scale settings.

**Questions:**

See Weaknesses.

---

### Official Review · Reviewer_aFzw · 2025-11-03

**Soundness:** 2
**Presentation:** 2
**Contribution:** 2
**Rating:** 4
**Confidence:** 4

**Summary:**

This paper identifies the implicit reliance on the pairing assumption of existing multimodal attribution methods. It extends Multimodal Information Bottleneck (M2IB) by dynamically controlling the tradeoff between compression and fitting.

**Strengths:**

- This paper studies how to dynamically control the tradeoff between compression and fitting, a common challenge for information bottleneck-based attribution methods.
- The experiments are comprehensive, covering relevant baselines and four different datasets

**Weaknesses:**

1. The implementation of f and g is unclear.
- What's the range of g? In definition 1, $g(f(X,Y)) \in (0,\inf)$. In sec 5.2, g is an MLP with relu (is relu added in the middle layers or after the last layer of mlp as in line 314?), which also doesn't have an upper bound.  However, in Figure 4, g seems to be bounded by [0,1].
- Is g learned for every sample? It seems that the authors use definition 2 in implementation, discarding the constraint that g is a nonincreasing function on f. If g is learned for each sample, as f(x,y) is a constant given x and y, we can keep maximizing the objective by pushing g(x,y) to 0.
- It's unclear why f is chosen to be a fixed function while g is a learnable function. The experiments have tested different fixed function choices for f and different architectures for g, but don't explain this core design.
- If f is intended to reflect the relevance between x and y, shouldn't f be inverse L2 instead of L2?
2. The authors state they prioritize I(Z;Y) for highly relevant image-text pairs, using L2 distance to determine this relevance. However, this metric is problematic for complex captions. A caption might list many objects present in the image alongside some that are not,  resulting in a higher L2 distance (and thus low "relevance"). In this scenario, all present objects should be highlighted, but the current implementation will encourage compression. Many visualizations presented in this paper have shown this over-compression. For example, in the "dog running down a snow covered hill" example (fifth row of figure 7), AdaIB highlights "dog" and "snow" "hill" in the text but only highlights the "dog" in the image. This suggests the adaptive mechanism may be too aggressive, pruning relevant semantic concepts that are part of the intended explanation.

**Questions:**

See weakness
- How's the AdaIB's performance on negation (e.g. "a photo without a dog")?
- How is Figure 1 AdaIB figure obtained? Each heatmap is usually normalized, so it should have at least some red areas. Is any thresholding applied?

---

### Note · Authors · 2025-12-01

I have read and agree with the venue's withdrawal policy on behalf of myself and my co-authors.